



# Atmospheric Boundary Layer height estimation from aerosol lidar: a new approach based on morphological image processing techniques

Gemine Vivone[1], Giuseppe D'Amico[1], Donato Summa[1], Simone Lolli[1], Aldo Amodeo[1], Daniele Bortoli[2,3], and Gelsomina Pappalardo[1]

[1]Consiglio Nazionale delle Ricerche, Istituto di Metodologie per l'Analisi Ambientale (CNR-IMAA), Tito Scalo, Potenza, Italy
[2]Institute of Earth Sciences (ICT), Pole of Evora, Evora, Portugal
[3]Physics Department, University of Evora, Evora, Portugal

**Correspondence:** Simone Lolli (simone.lolli@imaa.cnr.it)

**Abstract.** The Atmospheric Boundary Layer (ABL) represents the lowermost part of the atmosphere directly in contact with the Earth surface. The estimation of its depth is of crucial importance in meteorology and for anthropogenic pollution studies. The ABL Height (ABLH) measurements are usually far from being adequate, both spatially and temporally. Thus, different remote sensing sources can be of great help in growing both the spatial and temporal ABLH measurement capabilities. To this

aim, aerosol backscatter profiles are widely used as proxy to retrieve the ABLH. Hence, the scientific community is making remarkable efforts in developing automatic ABLH retrieval algorithms applied to lidar observations. In this paper, we propose a ABLH estimation algorithm based on image processing techniques applied to the composite image of the total attenuated backscatter coefficient. A pre-processing step is applied to the composite total backscatter image based on morphological filters to properly set-up and adjust the image to detect edges. As final step, the detected edges are post-processed through

both mathematical morphology and an object-based analysis. The performance of the proposed approach is assessed on real data acquired by two different lidar systems, deployed in Potenza (Italy) and Evora (Portugal), belonging to the EARLINET network. The proposed approach has shown higher performance than the benchmark consisting of some state-of-the-art ABLH estimation methods.

## 1 Introduction

The Atmospheric Boundary Layer (ABL) is the lowermost part of the atmosphere (50m-3000m) where the majority of human activity and meteorological phenomena are taking place. The Atmospheric Boundary Layer is the part of the troposphere that is directly or indirectly influenced by the Earth's surface (land and sea), and responds to gases and aerosol particles emitted at the Earth's surface and to surface forcing at time scales of less than one day. Forcing mechanisms include heat transfer, fluxes of momentum, frictional drag and terrain-induced flow modification (Bravo-Aranda et al., 2020). For this reason, the

ABL exhibits a strong diurnal variability depending on the solar cycle: stable, stratified and shallower at night; unstable, deeper, mixed and convective during strong solar irradiation (Arya, 1988; Mahrt, 1999). The ABL thermodynamic stability is of pivotal





importance in regulating turbulence, convection and precipitation, other than affecting the Earth-atmosphere exchanges of heat, momentum, pollutants and moisture (Mahrt, 1999; Su et al., 2020). Moreover, the underlying surface plays also a crucial role for the ABL development: the marine boundary layer follows completely different processes with respect to the boundary layer

over land, which in turn is influenced by the surface albedo because different surfaces respond differently to the solar heating (Sailor, 1995).

The ABL is crucial in meteorology and its depth norms the available volume that the anthropogenic pollutants emitted at surface can occupy, affecting their concentration and consequently the air-quality. For this reason, extremely bad air pollution episodes often happen during winter at extra-tropical latitudes, when a high-pressure anticyclone is making the ABL shallower

while an insufficient solar radiation is unable to trigger the convection and mixing. Instead, for tropical regions, the ABL depth is mostly related to the monsoon circulation (Canut et al., 2010; Lolli et al., 2019). For all the previously cited reasons, the ABL height (ABLH) has been recognized as a fundamental complex key meteorological variable to improve the aerosol dispersion forecast and re-analysis (Haeffelin et al., 2012), i.e., accurate and dense ABLH measurements are needed to tackle air-quality related issues in highly urbanized environment and greater metropolitan areas, being this variable directly related to pollutant

accumulation.

Despite its fundamental importance, ABLH measurements are far from being adequate, both spatially and temporally. The unique officially accepted measurements at global scale are carried out in the frame of the World Meteorological Organization (WMO) radiosounding global network. These measurements, highly concentrated within the most advanced Western countries in the Northern Hemisphere, are taken twice per day (00UTC and 1200UTC) without providing an adequate temporal coverage

neither capturing the ABL diurnal cycle due to the different time zone (Lolli et al., 2013; Cimini et al., 2020). The different remote sensing techniques might be of great help in growing both the spatial and temporal ABLH measurement capabilities. The thermodynamic atmospheric variables, i.e., the temperature and humidity, or the atmospheric profile of the wind speed, clouds and aerosols can be used as proxy to retrieve the ABLH from the vertically-resolved profiles taken by both active and passive remote sensing instruments as microwave radiometers, sodars, ceilometers and lidars. Those instruments are currently

deployed in the E-PROFILE network and the Aerosols, Clouds and Trace Gases Research Infrastructure (ACTRIS). The single-wavelength lidar, among all the different previously cited remote sensing techniques, is an active remote sensing instrument that provides vertically-resolved profiles of the optical and geometrical aerosol properties at high spatial and temporal resolution. The aerosol backscatter profile can be used as a tracer to retrieve the ABLH. The scientific community, in the past decades, made remarkable efforts in developing automatic ABLH retrieval algorithms applied to lidar observations. In fact, besides

the visual inspection (Boers et al., 1984) or simply setting up a threshold on the signal itself (Melfi et al., 1985), several algorithms existing in literature are based on detecting the abrupt change in aerosol concentration at the top of the boundary layer to retrieve the ABLH from each single lidar profile. Traditionally, some algorithms are based on detecting the strong gradient of the first derivative of the backscattered range corrected lidar signal profile (Flamant et al., 1997; Menut et al., 1999), while other ones are based on the second derivative (Sicard et al., 2006) or the first derivative of the logarithm (Summa

et al., 2013). Some alternative algorithms propose variants based on detecting the gradient of the normalized signal (He et al., 2006), of its cubic root (Yang et al., 2017) or fitting the lidar ideal profile (Steyn et al., 1999; Eresmaa et al., 2005). If the





cross-polarization channel measurement is available for the lidar instrument, the ABLH can be retrieved from the changes
in the lidar depolarization ratio as showed in (Bravo-Aranda et al., 2017). (Hooper and Eloranta, 1986) retrieved the ABLH
identifying the maximum variance of the backscattered signal, while in other several studies the ABLH is retrieved applying
the wavelet covariance transform (WCT) to the lidar signal (Cohn and Angevine, 2000; Davis et al., 2000; Talianu et al., 2006;
Compton et al., 2013). (Comerón et al., 2013) put in evidence that a strong link exists between the wavelet transform and the
gradient method. Other hybrid methods combine the wavelet approach with the image processing techniques (Morille et al.,
2007; Lewis et al., 2013) or pair the measured aerosol-backscatter lidar profile with models, e.g. multi-wavelength numerical
model (Dionisi et al., 2018); stability dependent model of ABLH temporal variation (Su et al., 2020). Recently, sophisticated
algorithms that include very advanced techniques were proposed: (De Bruine et al., 2017) developed an algorithm based on
tracking (pathfinder) that employs graph theory to track the diurnal evolution of the ABLH. (Toledo et al., 2014) applied for
the first time the Cluster Analysis (CA) to lidar measurements to obtain the ABLH. Another recent technology takes advantage
of an Ensemble Kalman Filter (EKF) to trace ABLH evolution from ground-based lidar observations as shown in (Tomás et al.,
2010). A more detailed review with more information about advantages and drawbacks of the reported algorithms can be found
in (Dang et al., 2019).

In this manuscript we present the development and validation of a new algorithm to continuously retrieve the ABLH from
measurements taken at different permanent observational sites deployed in the frame of the European Aerosol Lidar Network
(EARLINET) included in the ACTRIS research infrastructure (Pappalardo et al., 2014). The new implemented algorithm is
using a fully image-based methodology (2-D): instead of analyzing the lidar observations profile by profile, the retrieval takes
into account also the temporal dimension (at very high resolution). The algorithm framework needs as input a statistically
significant temporal collection of the total attenuated backscatter vertically-resolved profiles. The retrieval consists in applying
to the composite image during the pre-processing phase the morphological operators and the edge detectors. Instead, during
the post-processing phase, after applying the morphological filters again, the significant edges are extracted through an object-
based analysis that is proven to be particularly successful in determining the ABLH. This proof-of-concept is a useful starting
point to develop a central common strategy to produce high-quality and reliable ABLH retrieval in the frame of the Global
Atmosphere Watch (GAW) Aerosol Lidar Observation Network (GALION; (Bösenberg et al., 2008)) project of the WMO,
which has as main objective to harmonize all the existing lidar and ceilometer networks. Indeed, the proposed approach is able,
after a proper parameter tuning phase, to work with a huge variety of data addressing the task of the ABLH estimation even for
data acquired by simpler networks or less advanced systems.

The manuscript is organized as follows. Sect. 2 is devoted to the presentation of widely used state-of-the-art methods to
estimate the ABLH. In Sect. 3, the proposed image processing based approach is described. Afterwards, the experimental
results are shown in Sect. 4. Finally, conclusions are drawn in Sect. 5.



## 2   ABL Lidar Retrievals

This section is devoted to the description of widely used state-of-the-art techniques exploited to estimate the ABLH when lidar
data are involved using the aerosol backscatter profile as a tracer. In particular, several approaches can be used to this aim, as
detailed in Sect. 1. The most common used methods rely on gradient detection techniques or the use of the wavelet covariance
transform. In the following of this section, we will take into account these two main categories of algorithms that estimate
the ABLH with a special focus on the two methods belonging to the benchmark exploited to assess the performance of the
proposed morphological image-based approach.

### 2.1   Gradient-based for ABLH Estimation

There are several methods to determine the ABLH from lidar observations that are based on the assumption that aerosol is
trapped within the ABL. Those methods find the height where the aerosol concentration abruptly decreases. This happens
because the aerosol particles within the ABL can be used as proxy to study the boundary-layer vertical structure and time
variability. In fact, aerosols uplifted after sunrise by convective mixing can act as efficient tracers for the atmospheric portion
over which mixing occurs. Aerosols can also be dispersed out of the ABL during strong convective events or temporary breaks
of the capping temperature inversion. Thus, elastic backscattered signals from aerosol particles measured by lidar systems can
be used to determine the height and the internal structure of the ABL and, when possible, the residual layer and aerosol layers
within and aloft the ABL (Di Girolamo et al., 1999).

For lidar systems, typically the detected backscattered light is much higher within the ABL than in the free troposphere due
to the higher abundance of particles. The lidar equation is defined as:

$$P(\lambda, z) = P(\lambda_0, z) O(z) \frac{A}{z^2} \left[ \beta_{par}(\lambda, z) + \beta_{mol}(\lambda, z) \right] T_{mol}(\lambda, z) T_{par}(\lambda, z) + P_{bgd}(\lambda), \tag{1}$$

where $\lambda_0$ and $\lambda$ are the emitted and received lidar wavelength (laser wavelength), $z$ is the vertical height, $O(z)$ is the overlap
function, $A$ refers to a system function (area and typical configuration), $\beta_{mol}$ and $\beta_{par}$ are the backscatter coefficients for
molecular and particle components, respectively, $T_{mol}$ and $T_{par}$ indicate the atmospheric transmissivity and $P_{bgd}$ is the back-
ground signal. Often it is preferred to use the corrected signal for the square of the quota, named range corrected signal (RCS),
defined as:

$$P_{rcs}(\lambda, z) = \left[ P(\lambda, z) - P_{bgd}(\lambda) \right] z^2 \tag{2}$$

In order to determine an estimate of the height of ABLH, we directly apply the derivative method exploiting the logarithm
of the quantity (2). In this case, the lidar elastic backscatter signal, $P(\lambda, z)$, is used. Thus, we have that the ABLH, $H(\lambda)$, can
be obtained as follows:

$$H(\lambda) = \min_z \left\{ \frac{d}{dz} \log \left[ P_{rcs}(\lambda, z) \right] \right\}. \tag{3}$$

The minimum of the quantity in (3) identifies the transitions between different layers, and the absolute minimum identifies
the height of the boundary layer, because the largest variation of the lidar signal is considered corresponding to the largest





variation of the aerosol concentration. $H(\lambda)$ represents the transition from the stable layer to a neutral or unstable condition

above. The method has been applied to the maximum vertical spatial and temporal resolutions.

## 2.2 Wavelet Covariance Transform for ABLH Estimation

The Wavelet Covariance Transform (WCT) is defined as (Brooks, 2003)

$$c(b) = \frac{1}{a} \int\limits_{z_b}^{z_t} s(z)h\left(\frac{z-b}{a}\right) dz, \tag{4}$$

where $s(z)$ is the range-corrected lidar backscatter signal, $z$ is the measurement height, $z_b$ and $z_t$ are the lower and upper limits

of the lidar return signal profile, respectively, $a$ is the dilation factor, $b$ the translation, and $h$ is defined as the Haar wavelet

function, i.e.

$$h(x) = \begin{cases} 1, & \text{if } -\frac{1}{2} \le x \le 0 \\ -1, & \text{if } 0 < x \le \frac{1}{2} \\ 0, & \text{otherwise.} \end{cases} \tag{5}$$

In order to have a more efficient implementation and to gain more insights about WCT, (4) can be seen as convolution. Thus,

starting from (4) we have:

$$c(b) = \int\limits_{-\infty}^{+\infty} s(z)w(b-z)\, dz = s(b) * w(b) \tag{6}$$

where

$$w(x) = \frac{1}{a}h\left(-\frac{x}{a}\right), \tag{7}$$

$*$ stands for the convolution operator and $z_b$ and $z_t$ are set to $-\infty$ and $+\infty$, respectively, without harming the generality. It is

worth to be pointed out that the Haar function $h(x)$ in (5) can be rewritten using the derivative and a triangular window. Hence,

we have that:

$$h(x) = \frac{1}{2}\frac{d}{dx}\Lambda(2x), \tag{8}$$

where

$$\Lambda(x) = \begin{cases} 1+x, & \text{if } -1 \le x \le 0 \\ 1-x, & \text{if } 0 < x \le 1 \\ 0, & \text{otherwise} \end{cases} \tag{9}$$

is the triangular window.





Let us consider the Fourier transform of $c(b)$ in (6), where $b$ plays the role of the time domain in the classical Fourier analysis for linear time-invariant (LTI) systems. The convolution in time can be seen in the transformed domain as (Bracewell and Bracewell, 1986):

$$C(f) = S(f)W(f), \tag{10}$$

where $C(f) = \mathcal{F}[c(b)]$, $S(f) = \mathcal{F}[s(b)]$, $W(f) = \mathcal{F}[w(b)]$, and $\mathcal{F}[\cdot]$ is the forward Fourier transform. Considering the deriva-
tive and the scaling properties of the Fourier transform (Bracewell and Bracewell, 1986) and that the dilation $a \geq 0$, after simple algebra, starting from (7) and (8) we have that:

$$W(f) = \mathcal{F}[w(b)] = i\frac{\pi}{2}f\,sinc^2\left(\frac{a}{2}f\right) \tag{11}$$

where $i$ is the imaginary unit and

$$sinc(x) = \frac{\sin(\pi x)}{\pi x} \tag{12}$$

is the normalized sine cardinal function.

    Now if we consider the modulus of $C(f)$, i.e., $|C(f)|$, in (10), we have that $|C(f)| = |S(f)||W(f)|$, where the modulus of $W(f)$ is

$$|W(f)| = \frac{\pi}{2}|f|\,sinc^2\left(\frac{a}{2}f\right). \tag{13}$$

Thus, it is easy to see that if $f \to \pm\infty$, $|W(f)| \to 0$ and if $f = 0$, $|W(f)| = 0$. Indeed, the Fourier transform of the triangular
window in (8) can be seen as a low-pass filter with a cut-off depending on $a$ (i.e., the factor in (13) represented by the $sinc^2$ function). Instead, the derivative in (8) leads to a multiplication by $|f|$ in (13) generating a band-pass filter with cut-off frequencies depending on $a$, which rules the selection of a portion of the whole frequency spectrum. Hence, we have from (11) that higher frequencies of $s(b)$ are passed with respect to the low-pass filter defined by the triangular windows. In this sense, the WCT method with the Haar wavelet function can be considered as a particular gradient-based method.

In all the equations above, we neglected the dependence on $a$ of the WCT $c$. Indeed, the dilation $a$ is set to a fixed value. A rule of thumb can be found in (Baars et al., 2008). However, in this paper, in order to have a high performance method for our benchmark, we set $a$ to its optimal value (which is sensor-dependent) obtained via a grid search approach. Furthermore, following the indications in (Baars et al., 2008), we normalize the range-corrected signal by its maximum value found below a given height (usually set around 1000 m). The normalization guarantees the applicability of a unique threshold (set to 0.05 as
suggested in (Baars et al., 2008)) on $c(b)$ in order to find the ABL even at very different backscatter conditions in rather clean or very polluted air.

## 3    The Proposed Morphological Image Processing Approach (MIPA)

The composite image on which the proposed algorithm is applied consists in the temporal sequence of the range-corrected vertically-resolved acquired lidar profiles. The proposed MIPA algorithm has no prior knowledge on ABLH and the image



processing approach relies on: $i$) a block that reduces the vertical spatial resolution to reach a working spatial resolution around 20m if the spatial resolution of the system is finer; $ii$) a pre-processing step applied to the daily lidar data exploiting mathematical morphology; $iii$) Canny's edge detection (Canny, 1986) applied to the pre-processed data; $iv$) a post-processing starting from the detected edges and based on both mathematical morphology and an object-based analysis to get the final outcome. In the following sections, the basics of morphological operators are presented first. Then, the proposed MIPA algorithm

will be detailed.

### 3.1 Basics of Morphological Operators

An image $\mathbf{I}: E \subseteq \mathbb{Z}^2 \to V \subseteq \mathbb{Z}$ is analyzed by morphological operators via the so-called structuring element (SE), here denoted as $B$ (Soille, 2003), which can be defined through its spatial support $N_B(\mathbf{x})$ (i.e., the neighborhood with respect to the position $\mathbf{x} \in E$ in which $B$ is centered) and by its values. For *flat SEs* (i.e., SEs with unitary values), the only free parameters are the

origin and $N_B$. We will focus on these SEs in this work.

*Erosion*, $\varepsilon_B[\mathbf{I}]$, and *dilation*, $\delta_B[\mathbf{I}]$, are the two basic operators defined, for each $\mathbf{x} \in \mathbf{I}$, as follows:

$$\varepsilon_B[\mathbf{I}](\mathbf{x}) = \bigwedge_{\mathbf{y} \in N_B(\mathbf{x})} \mathbf{I}(\mathbf{y}), \tag{14}$$

$$\delta_B[\mathbf{I}](\mathbf{x}) = \bigvee_{\mathbf{y} \in N_B(\mathbf{x})} \mathbf{I}(\mathbf{y}), \tag{15}$$

where $\bigwedge_S$ and $\bigvee_S$ are the infimum and supremum values in the set $S$, respectively.

The erosion (respectively dilation) application has as filtering effect that suppresses bright (respectively dark) regions smaller than $B$ and the enlargement of dark (respectively bright) ones. For bright and dark regions we mean that the local contrast in a certain region has intensity values greater or lower with respect to the surrounding ones, respectively. Erosion and dilation operators can be recast into minimum and maximum operators on $B$, respectively, if $\mathbf{I}$ is a binary image.

We also introduce for convenience the *opening* and *closing* that correspond to the two possible sequential compositions of erosion and dilation. In particular, the *opening* is defined as:

$$\gamma_B[\mathbf{I}] = \delta_{\check{B}}[\varepsilon_B[\mathbf{I}]], \tag{16}$$

where $\check{B}$ denoting the SE obtained by reflecting $B$ with respect to its origin. Instead, the *closing* is given by

$$\phi_B[\mathbf{I}] = \varepsilon_{\check{B}}[\delta_B[\mathbf{I}]]. \tag{17}$$

A closing removes dark regions smaller than $B$, whereas an opening suppresses bright ones. For further details about morphological operators, the interesting readers can refer to the related literature (Soille, 2003).

A number of morphological operators can be obtained by properly combining the above-mentioned operators. Two instances, which are of great interest in this work, are represented by the residuals of the application of erosion and dilation, usually called



*internal gradient* and *external gradient*, respectively (Soille, 2003). In particular, the *internal gradient*, $\rho_B^-[\mathbf{I}]$, is defined as

$$\rho_B^-[\mathbf{I}] = \mathbf{I} - \varepsilon_B[\mathbf{I}], \tag{18}$$

and the *external gradient*, $\rho_B^+[\mathbf{I}]$, is given by

$$\rho_B^+[\mathbf{I}] = \delta_B[\mathbf{I}] - \mathbf{I}. \tag{19}$$

These two gradients are also often called *Half-Gradients* (HGs).

In the remaining of this section, the four modules that constitutes the proposed approach will be detailed.

### 3.2 Profile Resolution Reduction

This first block starts from a matrix $\mathbf{I}: E \subseteq \mathbb{Z}^2 \rightarrow V \subseteq \mathbb{Z}$ that is the daily sequence of the attenuated backscatter profiles. These
latter form the columns of $\mathbf{I}$. Thus, the number of rows is related to the maximum height and the spatial resolution of the
system, instead the number of columns is about its temporal resolution. The downsampling with a factor $R$, which aims to
reduce the bins' spatial resolution, is implemented by a low-pass filter along each column of $\mathbf{I}$ plus decimation with a factor $R$.
In particular, a moving-average filter is simply applied as low-pass filtering. The support (i.e., the length of the sliding window)
of the filter is $R$, again. Thus, $R$ is the unique tuning parameter that is selected in order to have a spatial resolution not finer
than 20 m. Hence, $R$ can be directly calculated from the system's spatial resolution bringing the initial product to the target
spatial resolution. This operation is performed to have a sharper edge defining the ABL. The outcome is an image $\mathbf{I}_D$ used as
input in the pre-processing step. It is worth to be remarked that if we work with data having a spatial resolution coarser than
20 m, $R$ is set to one, thus skipping this step implying that $\mathbf{I}_D$ is equal to $\mathbf{I}$.

### 3.3 Pre-processing Based on Mathematical Morphology

In this work, we propose the use of a low-pass filter based on HGs to pre-process $\mathbf{I}_D$. Before coming into the details of the used
operator, let us define the complement of a generic operator $\Psi$, *i.e.*, $\overline{\Psi}$, as $\overline{\Psi} = id - \Psi$, where $id$ is the identity operator. It is
worth to be remarked that when discontinuities are present both the HGs assume positive values constituting an approximation
of the norm of the signal gradient (Soille, 2003). Thus, the difference of the two (internal and external) HGs represents a detail
extraction operator, since it reproduces the variations of the function with respect to the local mean (Soille, 2003). This operator
exploiting a SE $B_{pre}$, denoted as $\overline{\Psi}_{HG,B_{pre}}$, is defined as follows:

$$\overline{\Psi}_{HG,B_{pre}} = 0.5\left(\rho^- - \rho^+\right) \tag{20}$$
$$= 0.5\left(id - \varepsilon_{B_{pre}}\right) - 0.5\left(\delta_{B_{pre}} - id\right), \tag{21}$$





in which the factor $0.5$ is applied to preserve the property of approximating the image gradient norm. The corresponding low-pass filter is simply given by the complementary operator of $\overline{\Psi}_{HG,B_{pre}}$, i.e.

$$\Psi_{HG,B_{pre}} = id - \overline{\Psi}_{HG,B_{pre}} \tag{22}$$
$$= id - \left[ 0.5(id - \varepsilon_{B_{pre}}) - 0.5(\delta_{B_{pre}} - id) \right] \tag{23}$$
$$= 0.5 \left( \varepsilon_{B_{pre}} + \delta_{B_{pre}} \right), \tag{24}$$

which corresponds to the semi-sum of the dilation and erosion operators. This operator is used in the pre-processing phase applying it to $\mathbf{I}_D$ and fixing $B_{pre}$ to a *line SE* in the horizontal direction (i.e., along the time direction) with a length $l_{pre}$. It enables us to smooth the lidar image along the horizontal axis (where the dynamic of the ABL is quite slow), directionally reducing the noise and preserving the vertical edges that will be of crucial importance for the next step. The resulting image after pre-processing the image $\mathbf{I}_D$ is indicated as $\mathbf{I}_{pre}$ and it represents the input of the edge detection block.

## 235  3.4   Edge Detection

This processing step can be implemented in several ways. Every edge detector can be exploited to extract a first estimation of the ABL starting from $\mathbf{I}_{pre}$. Thus, the proposed approach is flexible and this block can be changed to possibly improve the results. Approaches, already discussed in this paper, as Wavelet Covariance Transform (WCT) or gradient-based could be adopted. However, the analysis of the performance varying these strategies in the proposed framework is out-of-the-scope of this paper.

Thus, we employed Canny's edge detector (the classical version available in commercial software as MATLAB) (Canny, 1986) to get the first estimation of the edge map, denoted as $\mathbf{E}$. The detected edges in $\mathbf{E}$ are indicated with $1$, instead, the rest of the map (background) is labeled as $0$. All the bins labeled as $1$ in the edge map are potential candidates to represent the ABL.

### 3.5   Post-processing

After applying Canny's edge detector to the pre-processed data, the edge map $\mathbf{E}$ is analyzed by using further signal processing.

In particular, morphological filters are exploited first. Then, a final post-processing phase relied upon an object-based analysis is performed. The two steps will be detailed in the following.

#### 3.5.1   Post-processing Based on Morphological Operators

The post-processing based on morphological filters is applied to the edge map $\mathbf{E}$. This processing step is about removing unrealistic edges (i.e., edges that vary too fast with respect to the dynamic of the ABL). Thus, we apply a series of directional

Low-Pass (LP) morphological filters. In particular, the used filters are obtained by sequentially applying an opening and a closing operator using the same structuring element $B_{post}$, i.e.

$$\overline{\Psi}_{LP,B_{post}} = \phi_{B_{post}} \gamma_{B_{post}}, \tag{25}$$





where $B_{post}$ is a *line SE* with a length $l_{post}$ and an angle $\theta$. The application to $\mathbf{E}$ of these directional filters varying $\theta$ from $\theta_{min}$ to $\theta_{max}$ and combining the outputs with a maximum operator provides the output of this post-processing step, indicated as $\mathbf{E}_{post}$.

### 3.5.2 Object-based Post-processing

The object-based post-processing is applied to the edge map $\mathbf{E}_{post}$. The detected edges are indicated with 1, instead, the rest of the map (background) is labeled as 0. The first layer in $\mathbf{E}_{post}$ consists of the first (starting from the ground) bins detected as "edge" (i.e., labeled as 1) analyzing the edge map for each profile (*i.e.*, in the vertical direction).

We work on these edges extracting objects. The main concept is the use of the *connectivity* in an edge map, i.e., the way in which the bins labeled as "edge" (which assume value 1 in the edge map) are spatially related to their neighbors. A bin declared as "edge" is said "8-connected" if exists at least a bin belonging to its 8-neighborhood, i.e., the adjacent bins in vertical, horizontal, and diagonal directions, declared as "edge", as well. All the bins that are "8-connected" to each other form an object.

Thus, several objects, clustering the bins declared as "edge", are collected and analyzed. In particular, an analysis about the spatial variability of these objects is performed. Indeed, if the absolute Euclidean distance between the mean of the heights for each extracted object (using the connectivity procedure explained above) and the related mean calculated on the objects in its neighborhood exceeds a threshold $\delta_{post}$, this object is removed from the solution. Finally, the outcome, i.e. the estimated ABL denoted as $\mathbf{E}_{out}$, is obtained by linearly interpolating the remaining objects in the edge map.

### 3.6 Overview of the Proposed MIPA Algorithm

Algorithm 1 (MIPA) summarizes the sequence of the adopted signal processing steps in order to provide to the readers a complete overview of the proposed approach.

---
**Algorithm 1** The proposed ABL estimation algorithm.

---
- Reduce the profile resolution of $\mathbf{I}$ by a factor $R$ to get $\mathbf{I}_D$

- Pre-process $\mathbf{I}_D$ by low-pass filtering using HGs, see (24), as described in Sect. 3.3, to get $\mathbf{I}_{pre}$

- Estimate the edges of $\mathbf{I}_{pre}$ using Canny's edge detector obtaining the edge map $\mathbf{E}$

- Post-process the edge map $\mathbf{E}$, as described in Sect. 3.5, using directional morphological filters as in (25) and the object-based analysis as described in Sect. 3.5.2, in order to get $\mathbf{E}_{out}$

---

## 4 Results

This section describes the results obtained by applying the ABL detection algorithms detailed in the previous sections on several high-resolution total attenuated backscatter lidar timeseries. In particular, as we use the aerosol as proxy to determine the ABLH, we considered observations at a longer wavelength (1064 nm) to get a higher contrast among the particle and molecular





contribution. In order to show the robustness of the proposed methodology, we selected different case studies characterized by different atmospheric conditions in terms of both aerosol loading and solar background. Further, we applied MIPA on observations from two quite different multi-wavelength sensors: the Potenza lidar system MUSA and the Evora lidar system PAOLI.


MUSA is one of the reference lidar systems in the frame of EARLINET deployed at CNR-IMAA Atmospheric Observatory (CIAO) in Potenza (40.60N, 15.72E, 760 m asl). The lidar instrument is equipped with 3 elastic channels at 355, 532 and 1064 nm and 2 anelastic N2 Raman channels at 387 and 607 nm (Madonna et al., 2011). Two independent polarization components of the elastic channel at 532 nm are separately detected in order to measure the particle depolarization ratio. All channels have a raw vertical resolution of 3.75 m and, except for the 1064 nm where only analog detection is used, all the channels are acquired both in analog and photoncounting mode to enhance the detectable dynamic range. The typical raw time resolution is 60 s.


PAOLI is the Evora EARLINET lidar system (38.57N, -7.91E, 293 m asl) and it operates 3 elastic channels (at 355, 532 and 1064 nm) and two anelastic channels at 387 and 607 nm. The total and the cross polarization channel are separately detected in the green. Only photoncounting detection mode is used for all the channels. The raw vertical resolution is 30 m and the time resolution is 30 s.


Even if MUSA and PAOLI operate at the same wavelengths, their technical characteristics are very different: laser source, telescope, detection and acquisition system, space and time resolution, full overlap region. As a consequence, applying MIPA algorithm on both systems is a good benchmark to evaluate the algorithm performances.


MIPA algorithm uses as input the composite plot of the vertically-resolved attenuated backscattering coefficient at 1064 nm. High spatial and temporal resolution is needed to increase sensibility in using the directional morphological filter while long and continuous time series are needed to improve the accuracy in mapping the detected edges. The proposed case studies are continuous multi-day observations from MUSA and PAOLI lidar systems. in July 2012, during the EARLINET 72 hours exercise (Sicard et al., 2015), both Evora and Potenza EARLINET lidar stations performed continuous measurements during


72 hours as proof of concept to demonstrate that EARLINET lidar network can provide aerosol optical products in NRT as operational service. The observations were automatically analyzed in NRT by the EARLINET Single Calculus Chain (SCC) (D'Amico et al., 2015), a common algorithm developed to centrally analyze and retrieve aerosol optical, geometrical and microphysical properties from the different EARLINET lidar instruments. The results obtained by applying the ABLH detection algorithm on both Potenza and Evora 72h datasets are described in Sect. 4.1. Additionally, we selected further 3 Potenza cases


described in Sect. 4.2.

The total attenuated backscatter can be easily expressed in terms of measured lidar signals taking into account (1) and (2):

$$\beta_{att}(\lambda, z) = [\beta_{mol}(\lambda, z) + \beta_{par}(\lambda, z)] T_{mol}^2(\lambda, z) T_{par}^2(\lambda, z) = K P_{rcs}(\lambda, z) \tag{26}$$

where $K$ is a calibration constant determined by imposing that $\beta_{att}(\lambda, z) = \beta_{mol}(\lambda, z) T_{mol}^2(\lambda, z)$ in an aerosol-free atmospheric region. It is important to note that morphological filter techniques rely only on the correlation among adjacent lidar


range bin and not on their absolute numeric values. As a consequence, the ABLHs computed by the proposed MIPA algorithm





are rather insensitive to the accuracy of the calibration constant $K$. Actually, the morphological algorithm can be applied to the range corrected signal ($P_{rcs}$) time series instead of the total attenuated backscatter one, providing the same results in terms of ABLH. This is in general not true for the traditional ABLH retrieval algorithms (derivative, WCT) where proper thresholds on absolute signal value need to be defined. Tab. 1 summarizes all the input parameters exploited by the compared approaches

for the two lidar systems considered. These have been defined after a tuning phase on different lidar scenarios. It is worth to be remarked that the derivative approach requires no parameter to be set working at the higher resolutions possible.

All the input data sets considered in the study have been previously pre-processed at high resolution by using the EARLINET SCC. In particular, starting from raw lidar data, several instrumental corrections (like for example trigger delay correction, dead time correction, analog and photoncounting signal glueing, etc.) and all the required raw signal handlings (like atmospheric

background subtraction, range correction) have been applied. More details on the pre-processing procedure implemented in the EARLINET SCC are described in (D'Amico et al., 2016).

To asses the performance of all the considered ABLH retrievals, a reference needs to be set. As the most rigorous definition of the PBL is the one based on thermodynamic effects, we assume as reference for the ABLH the values retrieved from atmospheric temperature and pressure profiles of the co-located radiosondes and the ECMWF model (if available) related to

positions closest to the site of the lidar system under study. The lidar retrieval uses the aerosols as tracers identifying the height of the maximum vertical gradient indicating the drop in aerosol concentration. This corresponds to that part of the atmosphere where convection and turbulence drops and we get the ABLH to be used as reference for the approaches estimating the ABL from lidar data (Summa et al., 2013).

The sounding data available for all the measurement dates considered in this study are quite few and except from one

case they are not co-located with the lidar station. In particular, there are no sounding data available for Potenza for any of the 72h exercise day. The closest sounding stations are Brindisi (40.66N, 17.96E, 15 m asl and located about 180 km away from Potenza EARLINET station) and Pratica di Mare (41.67N, 12.45E, 32 m asl located about 300 km away from Potenza EARLINET station) which are both coastal sites with very different atmospheric conditions with respect to the mountain Potenza lidar station (760 m asl). As a consequence, the corresponding temperature and pressure profiles cannot be used to

retrieve a reliable ABLH reference. Only for one of the additional selected Potenza measurement case (November 20, 2014), a single radiosounding launched at CIAO observatory (Madonna et al., 2011) is available, which however provides only one reference point that is not enough to well assess the performances of the lidar-based ABLH retrieval. For Evora EARLINET observational site, the closest sounding station is Lisboa/Gago Coutinho (38.77N, -9.13E, 110 m asl and located about 110 km away from Evora EARLINET station) at noon local time daily. Even if in this case the atmospheric conditions characterizing

Lisboa/Gago Coutinho sounding station could be considered similar to the ones characterizing the Evora EARLINET station, the 72h exercise ABLH reference points obtainable by using close soundings are only three and all referring to the same hour of the day.

We have used the atmospheric temperature and pressure profiles provided by the Numerical Weather Prediction (NWP) model over the two measurement sites for assessing the performance. Even though sounding data can be available sometimes

(as reported in this work when available for the scenarios under test), these data are not enough to guarantee the calcula-





tion of a reliable reference for the considered ABL retrievals due to the very low temporal resolution of these acquisitions. To our knowledge, NWP is the best alternative to the use of sounding data. In particular, to calculate the ABLH reference points for all the Potenza cases, we have used the high-resolution NWP provided by the ECMWF Integrated Forecast System (https://www.ecmwf.int/en/forecasts). We have used forecasts providing temperature and pressure profiles in correspondence

of 91 model levels on a $0.1° \times 0.1°$ grid. The vertical resolution increases linearly starting from 20-30 m up to about 400 m below 6 km allowing quite accurate determination of the ABLH. Moreover, the forecast time resolution of 1 h ensuring the calculation of sufficient number of ABLH reference points for all the considered Potenza cases. The ABLH reference points for the Evora 72h exercise dataset has been calculated using the forecasts provided by Global Data Assimilation System (GDAS) of the National Centers of Environmental Predictions of NOAA (https://www.ready.noaa.gov/gdas1.php). In this case the at-

mospheric temperature and pressure profiles are given for 23 model levels on a $1° \times 1°$ grid. The vertical resolution increases with the altitude starting from values of 200-300 m close to ground and reaching values of about 800 m at around 6 km. The forecast time resolution is 3 h. We used GDAS forecasts to calculate the ABLH reference for 72h Evora dataset because we do not have access to the corresponding ECWMF NWP. Both ECMWF and GDAS forecasts have been obtained through the CloudNET data portal (http://devcloudnet.fmi.fi).

## 360   4.1   EARLINET 72h Exercise

Fig. 1 shows the total attenuated backscatter time series at 1064 nm measured by the MUSA system during the 72h EARLINET exercise. The lidar observations started on July 9, 2012 at 08:00UTC and went on almost continuously until July 12, 2012 08:00UTC. The aerosol load observed during these measurements is consistent with a typical summer day in Potenza. The aerosol aloft is mainly dust while the layer at surface (into the ABL) is typically composed by local aerosols mixed with dust.

Often, the aerosols in the free troposphere tend to intrude in the ABL making the separation between the ABL and the upper atmospheric region not so clear. This is clearly visible around noon on 10 July and 11 July 2012. The general idea behind the retrieval of ABLH from lidar measurements is to use the aerosol as tracers of the ABL and assuming that the ABLH is given by the top of the first aerosol layer (starting from the surface). According to this assumption, we can clearly see that the ABLH is maximum around noon (when the solar convention is at its maximum) for all the three days and it reaches its minimum

during nighttime. The black line in Fig. 1 shows the ABLH as retrieved by MIPA algorithm. In general, the expected temporal evolution of the ABLH is well captured and the intrusions of the upper aerosol layers in the ABL do not seem to affect the outcomes, thus still obtaining reasonable ABLH estimates.

It is important to highlight that the assumption to retrieve the ABLH as the top of the first detected aerosol layer is valid only if the ABL is above the full lidar overlap height. During night-time observations, this condition may be not always verified

and in this case the algorithm detects as first edge the base of the layer next to the ABL top. This is clearly visible during the nighttime period in Fig. 1 where the real ABL is too low to be detected with the MUSA system and the retrieved ABLH is typically overestimated (Di Girolamo et al., 1999).

As explained in Sect. 3, the MIPA algorithm is composed by four different modules: an edge detector based on Canny filter (see Sect. 3.4), a module for reducing the vertical resolution (see Sect. 3.2), the pre-processing module described in Sect. 3.3





to be applied before the edge detector and, finally, a post-processing (see Sect. 3.5) to be applied after the edge detector step. Fig. 2 shows the role played by each of these modules in retrieving the ABLH for the Potenza 72h dataset and also the reference ABLH retrieved using the ECMWF forecasts (brown circles). The ABLHs shown in Fig. 2 were calculated considering the different steps of the MIPA algorithm, i.e. applying: the edge detector module on full resolution data (curve $a$ in Fig. 2), the vertical resolution reduction module and the edge detector (curve $b$), the vertical resolution reduction module, the pre-

processing module and the edge detector (curve $c$) and finally the whole algorithm (curve $d$). It is evident that the edge detector is not sufficient to ensure a stable retrieval even if it is applied to a reduced resolution dataset. Consequently, the application of the post-processing module is crucial in the processing. The absolute differences of all the curves shown in Fig. 2 with respect to the reference are plotted in Fig. 3. In generating Fig. 3 (and all other figures showing the absolute difference with respect to the reference), the reference data have been interpolated at the same resolution of the lidar data assuming that the ABL is

slowly varying.

Fig. 4 shows the comparison between the ABLH retrieved by the MIPA algorithm and the corresponding one obtained by using more traditional ABLH detection approaches. Specifically, we applied to the same dataset different algorithms for the ABLH estimation: the MIPA algorithm 1, the derivative method described in Sect. 2.1 and the procedure based on WCT as in Sect. 2.2. The ABLH retrieved using the ECMWF forecasts is reported as brown circles in Fig. 4. The agreement among the

ABLH obtained by applying the considered algorithms on lidar data is satisfactory. The MIPA algorithm provides the smallest discrepancies with respect to the reference. As expected, all the algorithms overestimate the ABLH in nighttime conditions where the real ABLH as retrieved by ECMWF forecasts data is clearly below the full overlap of the MUSA lidar, which is about 300 m.

The absolute difference of all the considered algorithms with respect to the reference is plotted in Fig. 5 and a statistical

analysis is reported in Tab. 2 for all the considered algorithms. Both Fig. 5 and Tab. 2 confirm the better performance of the MIPA algorithm with respect to the other approaches into the proposed benchmark.

Fig. 6 reports the lidar observations at 1064 nm made by PAOLI over the Evora site during the 72h EARLINET exercise. Differently from Potenza, there are not lofted aerosol layers in the atmosphere and most of the aerosol load is trapped in the ABL. The typical evolution of the ABL due to solar activity is clearly visible in the plot. Moreover, sometime there are small

convective clouds formed on the top of ABL (see for example around 6:00 UTC of July 11). The overlap of the PAOLI system is around 750-800 m. The MIPA algorithm captures quite well the evolution of the ABLH (black line in Fig. 6).

Figs. 7 and 8 are similar to Figs. 4 and 5, respectively, but refer to the 72h Evora dataset. The only difference with respect to Potenza 72h dataset is that, in this case, the reference is calculated exploiting atmospheric temperature and pressure profiles provided by the GDAS forecasts. Moreover, in Fig. 7, the ABLH retrieved by the soundings launched at noon of each days

from the Lisboa/Gago Coutinho sounding station (38.77N, -9.13E, 104.0 m asl and located about 110 km away from Evora EARLINET station) are also reported as green triangles. The very good agreement with the corresponding ABLH retrieved by using the GDAS forecasts proves the good accuracy of the selected reference.

Tab. 3 sums up the statistical analysis of the absolute differences with respect to the reference for all the considered algorithms for the 72h Evora dataset. As for Potenza, the better performance is the one corresponding to the proposed ABLH





retrieval algorithm. It is worth to be underlined that during daytime there is quite a good agreement between the GDAS fore-
casts and the outcomes of the MIPA approach, instead, during nighttime, the MIPA retrievals may be hindered by PAOLI's
overlap for low ABLHs.

## 4.2  Other Potenza Cases

In this section we describe three additional case studies on which the proposed MIPA algorithm has been evaluated. The three
cases refer all to MUSA lidar observations during three longer continuous observations ever made over Potenza with MUSA.

The first case study refers to April 21, 2010 when the Potenza EARLINET station performed a quite long record of lidar
measurement concurrently with the eruption of the Icelandic volcano Eyjafjallajökull occurred in April-May 2010. During the
eruption almost all the EARLINET stations promptly started continuous measurements, whenever weather conditions allowed
it and the corresponding outcomes are described by (Pappalardo et al., 2013).

Fig. 9 shows the comparison between the ABLH retrieved by the MIPA algorithm and the corresponding ABLH obtained
by using the other considered algorithms. ABLHs retrieved from ECMWF runs are assumed as reference (brown circles). The
absolute difference of all the considered algorithms with respect to the reference is plotted in Fig. 10. For this case, from
08:30UTC to 17:00UTC, all the considered lidar retrieval algorithms give ABLH values systematically below the reference
ones. This behavior can be explained looking at Fig. 11, where the total attenuated backscatter time series at 1064 nm measured
by the MUSA system on April 21, 2010 is shown. Moreover, the black line shows the ABLH retrieved by MIPA and the white
line shows the reference ABLH. Before the 8:30UTC the real ABLH is below the MUSA overlap and, consequently, as already
mentioned earlier, the proposed algorithm overestimates the ABLH capturing the edge of the next layer. Starting from the
08:30UTC, the solar activity initiates and the ABLH starts to increase above altitudes exceeding the lidar overlap. Thus,
from this point on, the ABLH retrieved by using lidar measurements should agree with the reference one. However, in this
particular case, the lidar observations show that two separated aerosol layers are present below the reference ABLH (about
3 km asl) indicating probably a not well mixed ABL. The bottom layer (below about 1.5-2.0 km asl) is probably composed
by local particles while the upper one contains dust of mixed dust. Clearly, in conditions like this, the ABLH retrieved by
lidar measurements (independently from the specific algorithm) will underestimate the real ABLH capturing the top of the first
aerosol layer and not the top of the ABL.

The other two cases refer to classic nighttime observations taken on Nov 20, 2014 and July 13, 2015. Figs. 12 and 14 report
the comparison of PBHL retrieved by all the considered algorithms for the selected cases, respectively. Additionally, Fig. 12
shows also the ABLH retrieved by using temperature and pressure profiles taken from a correlative radiosounding launched at
CIAO observatory (green triangle). The absolute differences for these two cases are shown in Figs. 13 and 15, respectively.

The agreement among all the considered algorithms is in general good for all the three cases. For all the cases in the dataset,
MIPA algorithm shows the highest accuracy with respect to the reference. This is confirmed also by the mean and median
values of the absolute difference with respect to the reference summarized in Tab. 4 (minimum values of both these parameters
are always obtained by using the proposed algorithm).





## 5  Conclusions

The estimation of the ABLH is of crucial importance both for meteorological and air-pollution related applications. In this
work, we proposed a new algorithm to continuously retrieve the ABLH. This approach leverages on the use of a fully image-
based methodology (instead of analyzing the lidar observations profile by profile). The retrieval consists in applying to the
image, during the pre-processing phase, morphological operators. Afterwards, an edge detection is considered. Finally, during
the post-processing phase, the significant edges are extracted through a further filtering phase based on mathematical morphol-
ogy and an object-based analysis. This approach has been compared with a proper benchmark consisting of state-of-the-art
ABLH estimation methods, i.e., a gradient-based approach and a WCT-based method. For the latter, the filtering capabili-
ties of the approach were pointed out. Different datasets acquired by two lidar systems located in two separated EARLINET
permanent observational sites have been considered to assess the performance.

The results, relying upon several statistical indexes, put in evidence that the proposed approach is more accurate than the
compared approaches belonging to the benchmark. In particular, we observed an improvement of the accuracy of about 30%
(on average) with respect to the closest state-of-the-art approach (i.e., the WCT). Moreover, the outcomes obtained by the
MIPA are more stable than the other benchmarking methods. This can be easily pointed out by having a look at the results
depicted in this paper and it has also been corroborated by calculating measures of dispersion (e.g., the standard deviation)
in the statistical analysis. The last concluding remark is about the computation analysis. Despite of the proposed approach
seems quite complex, it leverages on the use of very efficient filters based on mathematical morphology. The running times
on large datasets (72 hours) show excellent performance from this point of view requiring just few seconds for the execution
of the whole signal processing chain. The bottleneck of the system turns out to be the object analysis phase. However, the
computation times can be considered comparable with the other approaches proposed into the benchmark.

Finally, it is worth to be stressed that the MIPA approach does not depend on the absolute calibrated values but rather only
on the correlation among adjacent lidar range bins, which is of crucial importance in order to form the image. This interesting
feature could be very useful for future developments. Indeed, the MIPA approach could be easily adapted to address the task
of the estimation of the ABLH using other widely available and continuously acquired data, such as, ceilometer data.

*Author contributions.* Conceptualization, G.V. and S.L.; methodology, G.V.; software, G.V. and D.S.; validation, G.V., G.D.A., and D.S.;
formal analysis, G.V.; data curation, G.D.A., A.A., and D.B.; writing–original draft preparation, G.V., G.D.A., D.S., S.L.; writing–review
and editing, A.A., D.B., and G.P.; supervision, G.D.A. and G.P.; project administration, G.P.; funding acquisition, G.P.

*Competing interests.* The authors declare that they have no conflict of interest.



*Acknowledgements.* "ACTRIS (www.actris.eu) has received funding from the European Union's Horizon 2020 research and innovation programme under grant agreement numbers: 654109 (ACTRIS-2), 759530 (ACTRIS-PPP), 871115 (ACTRIS-IMP), 824068 (ENVRI-FAIR), and previously from the European Union Seventh Framework Programme (FP7/2007-2013) under grant agreement n° 262254.". Moreover, the authors gratefully acknowledge CloudNET for providing ECWMF and GDAS atmospheric forecasts for all the measurement cases

included in this study."



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



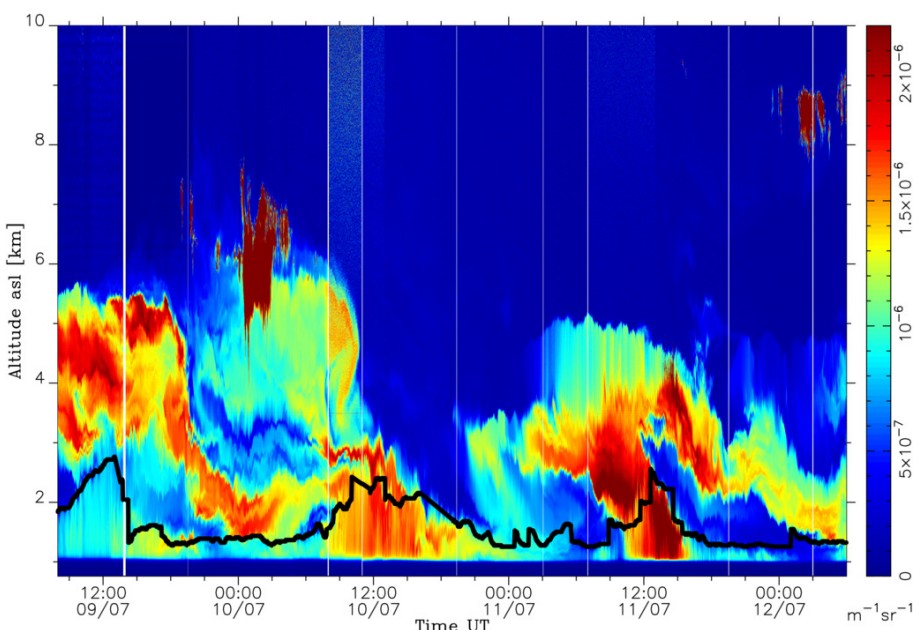

**Figure 1.** High-resolution time series of the total attenuated backscatter at 1064 nm measured over Potenza during the EARLINET 72h exercise (July 9-12, 2012). Time resolution is 60 s, vertical resolution is 3.75 m. The black curve shows the ABLH as retrieved by the proposed MIPA algorithm.

**Table 1.** Parameter setting for the WCT and MIPA approaches. The values of the parameters can be different for different lidar systems. Thus, they are reported for both MUSA (Potenza) and PAOLI (Evora) lidar systems.

| Parameter | MUSA | PAOLI |
|---|---|---|
| **WCT** | | |
| $a$ | 46 | 16 |
| **MIPA** | | |
| $R$ | 6 | 1 |
| $l_{pre}$ | 3 | 6 |
| $l_{post}$ | 6 | 6 |
| $\theta_{min}$ | -66° | -46° |
| $\theta_{max}$ | 66° | 46° |
| $\delta_{post}$ | 10 | 10 |





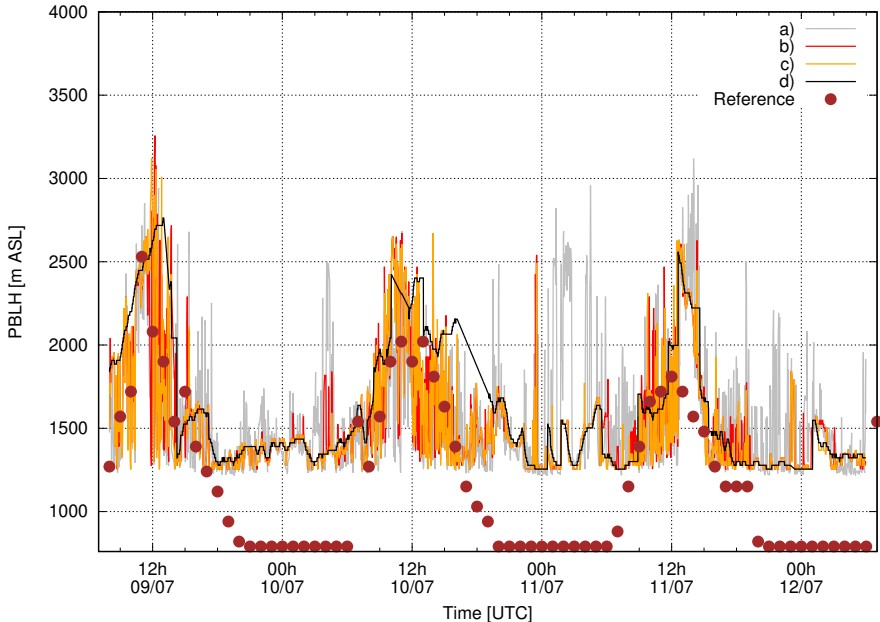

**Figure 2.** ABLH retrieved during the 72h EARLINET exercise (July 9-12, 2012) for Potenza. Gray, red, yellow and black lines show the ABLH retrieved from 1064 nm high-resolution total attenuated backscatter time series by applying the edge detector module on full resolution data (curve $a$), the vertical resolution reduction module and the edge detector (curve $b$), the vertical resolution reduction module, the pre-processing module and the edge detector (curve $c$) and the whole MIPA procedure (curve $d$). Brown circles represent the reference ABLHs as retrieved from atmospheric temperature and pressure profiles provided by ECMWF forecasts.

**Table 2.** Statistical analysis of the absolute differences with respect to the reference of the ABLH retrieved by applying MIPA, WCT and derivative approaches on Potenza 72h high-resolution time series of the total attenuated backscatter at 1064 nm. The mean ($\Delta_{mean}$), the median ($\Delta_{mean}$), the standard deviation ($\Delta_{std}$), the standard error ($\Delta_{ste}$), the minimum and the maximum of the absolute differences are given in meters. $N$ is the number of points on which the statistics are made. The reference is assumed to be the ABLH calculated from the co-located atmospheric temperature and pressure profiles provided by ECMWF forecasts.

|  | MIPA | WCT | Derivative |
|---|---|---|---|
| $\Delta_{mean}$ | 455 | 620 | 694 |
| $\Delta_{med}$ | 499 | 540 | 631 |
| $\Delta_{std}$ | 218 | 376 | 450 |
| $\Delta_{ste}$ | 26 | 45 | 54 |
| $\Delta_{min}$ | 25 | 62 | 21 |
| $\Delta_{max}$ | 875 | 1769 | 1719 |
| $N$ | 69 | 69 | 69 |





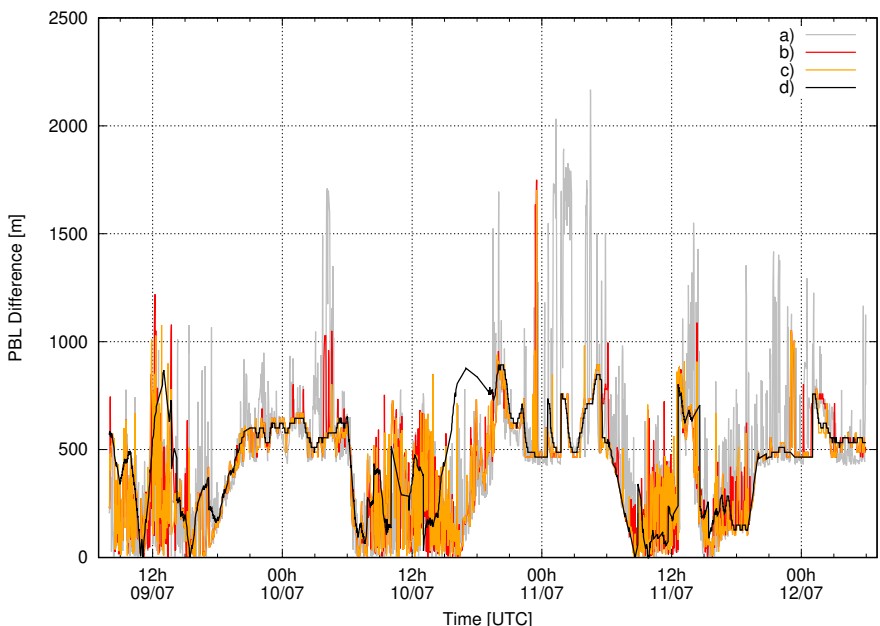

**Figure 3.** Absolute differences between the ABLHs retrieved from Potenza 72h high-resolution total attenuated backscatter time series at 1064 nm and corresponding ABLHs derived from atmospheric temperature and pressure profiles as provided by ECMWF forecasts. ABLHs retrieved by using ECMWF forecasts data are assumed to be the reference. The original time resolution of ECMWF forecasts is 1 hour. They have been interpolated at the lidar data time resolution (1 minute). Gray, red, yellow and black lines show the absolute differences between the curves a), b), c) and d) shown in Fig. 2 and the reference, respectively.

**Table 3.** Statistical analysis of the absolute differences with respect to the reference of the ABLH retrieved by applying MIPA, WCT and derivative approaches on Evora 72h high-resolution time series of the total attenuated backscatter at 1064 nm. The mean ($\Delta_{mean}$), the median ($\Delta_{mean}$), the standard deviation ($\Delta_{std}$), the standard error ($\Delta_{ste}$), the minimum and the maximum of the absolute differences are given in meters. $N$ is the number of points on which the statistics are made. The reference is assumed to be the ABLH calculated from the co-located atmospheric temperature and pressure profiles provided by GDAS forecasts.

|  | MIPA | WCT | Derivative |
|---|---|---|---|
| $\Delta_{mean}$ | 486 | 615 | 702 |
| $\Delta_{med}$ | 474 | 603 | 751 |
| $\Delta_{std}$ | 234 | 309 | 361 |
| $\Delta_{ste}$ | 49 | 64 | 75 |
| $\Delta_{min}$ | 80 | 78 | 159 |
| $\Delta_{max}$ | 962 | 1282 | 1455 |
| $N$ | 23 | 23 | 23 |



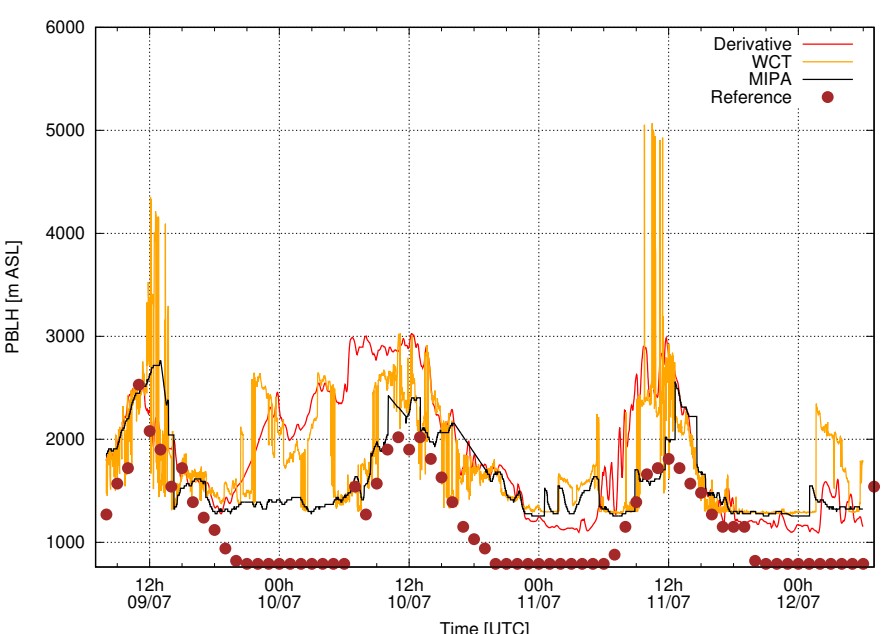

**Figure 4.** ABLH retrieved during the 72h EARLINET exercise (July 9-12, 2012) for Potenza. Black, red and yellow lines show the ABLH retrieved from 1064 nm high-resolution total attenuated backscatter time series applying MIPA, derivative and WCT algorithms, respectively. Brown circles represent the reference ABLH as retrieved from atmospheric temperature and pressure profiles provided by ECMWF forecasts.



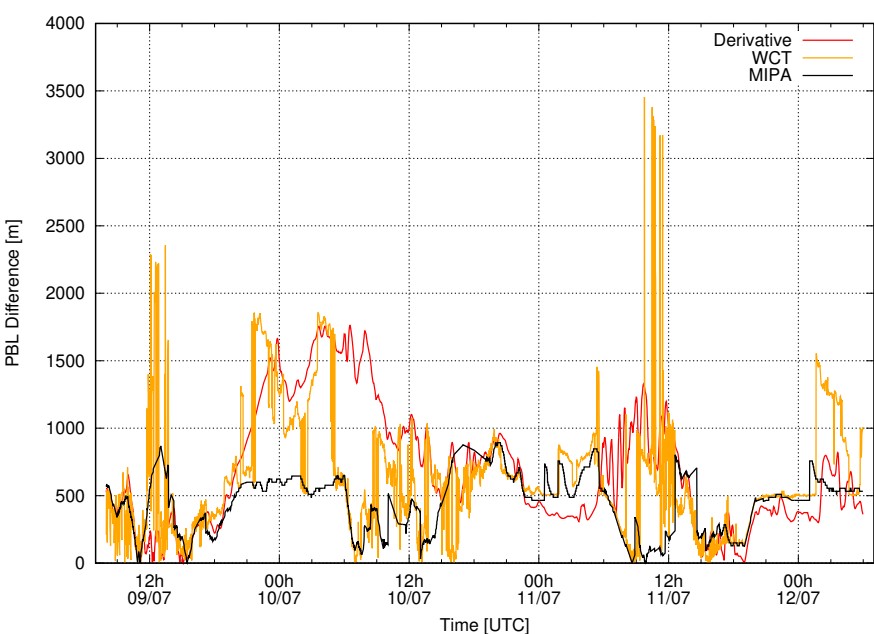

**Figure 5.** Absolute differences between the ABLHs retrieved from Potenza 72h high-resolution total attenuated backscatter time series at 1064 nm and the corresponding ABLHs derived from atmospheric temperature and pressure profiles as provided by ECMWF forecasts. ABLHs retrieved by using ECMWF forecasts data are assumed to be the reference. The original time resolution of ECMWF forecasts data is 1 hour. They have been interpolated at the lidar data time resolution (1 minute). Black, red and yellow lines show the absolute differences with respect to the reference of the ABLHs retrieved by MIPA, derivative and WCT algorithm, respectively.





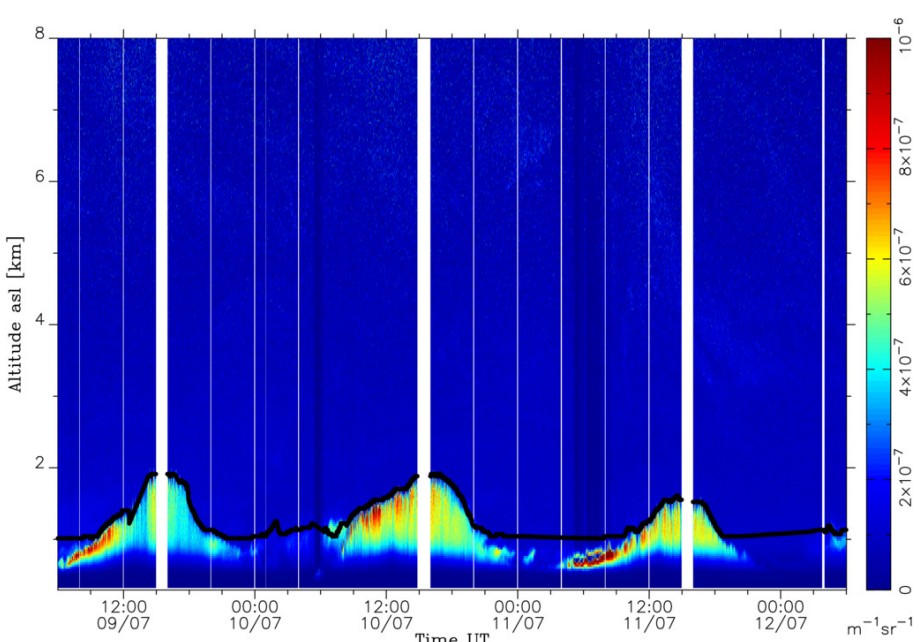

**Figure 6.** High-resolution time series of the total attenuated backscatter at 1064 nm measured over Evora during the EARLINET 72h exercise (July 9-12, 2012). Time resolution is 30 s, vertical resolution is 30 m. The black curve shows the ABLH as retrieved by the MIPA algorithm.

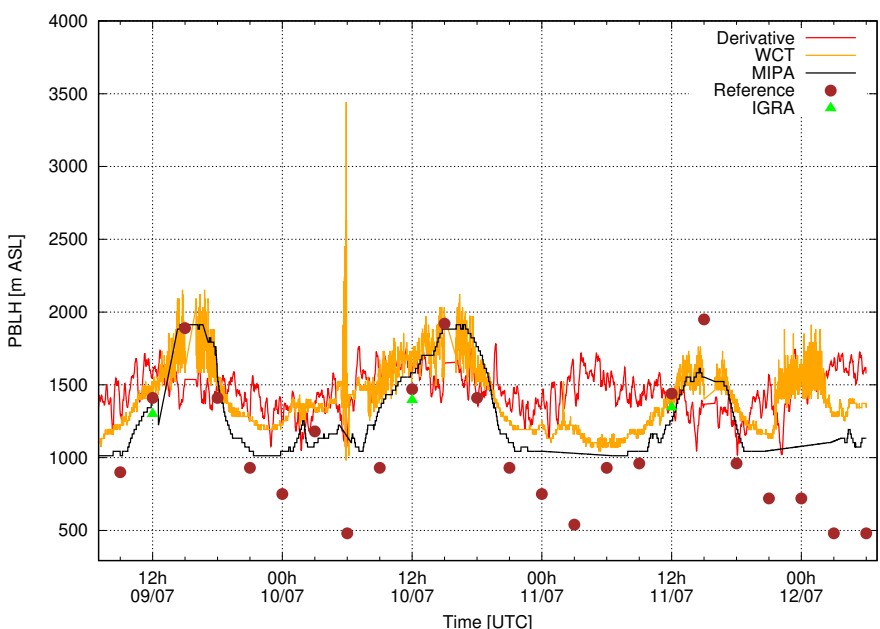

**Figure 7.** ABLH retrieved during the 72h EARLINET exercise (July 9-12, 2012) for Evora. Black, red and yellow lines show the ABLH retrieved from 1064 nm high-resolution total attenuated backscatter time series applying MIPA, derivative and WCT algorithms, respectively. Brown circles and green triangles are the ABLHs retrieved from atmospheric temperature and pressure profiles provided by GDAS forecasts and Lisboa sounding station (taken from Integrated Global Radiosounding Archive - IGRA), respectively.

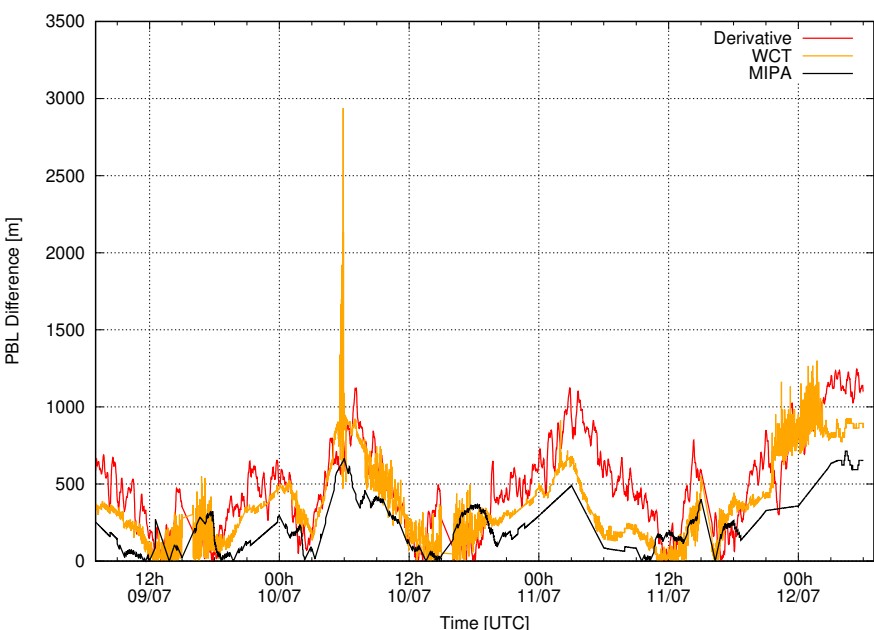

**Figure 8.** Absolute differences between the ABLHs retrieved from Evora 72h high-resolution total attenuated backscatter time series at 1064 nm and the corresponding ABLHs derived from atmospheric temperature and pressure profiles as provided by GDAS forecasts. ABLHs retrieved by using GDAS forecasts data are assumed to be the reference. The original time resolution of GDAS forecasts data is 3 hour. They have been interpolated at the lidar data time resolution (1 minute). Black, red and yellow lines show the absolute differences with respect to the reference of the ABLHs retrieved by MIPA, derivative and WCT algorithm respectively.



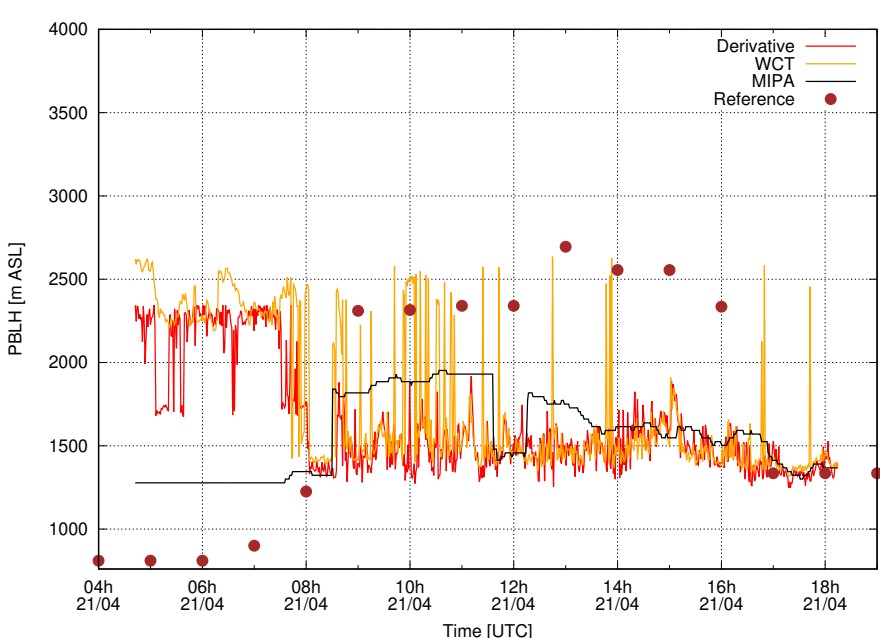

**Figure 9.** ABLH retrieved from the Potenza lidar measurements on April 21, 2010. Black, red and yellow lines show the ABLH retrieved from 1064 nm high-resolution total attenuated backscatter time series applying MIPA algorithm, derivative and WCT algorithms, respectively. Brown circles represent the ABLHs as retrieved from atmospheric temperature and pressure profiles provided by ECMWF forecasts.



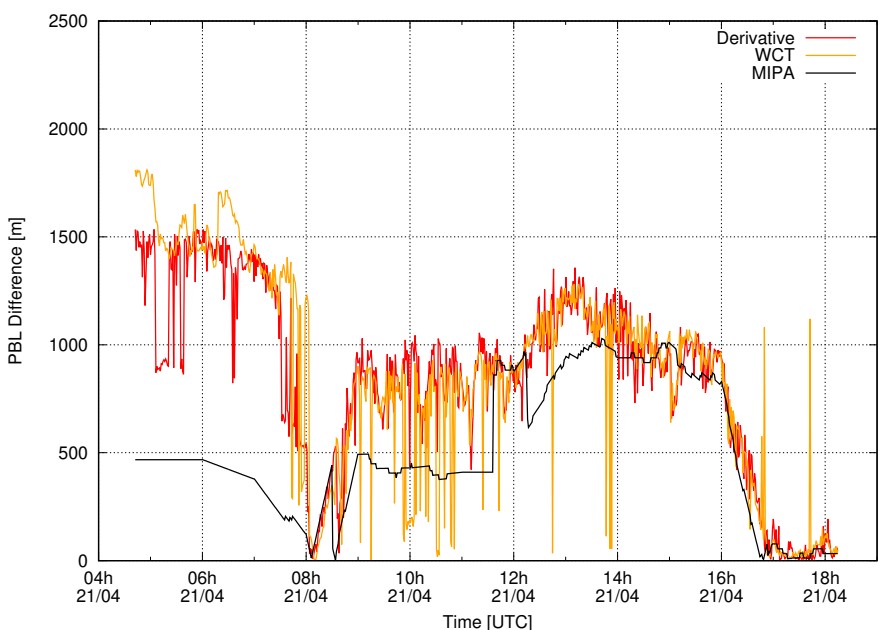

**Figure 10.** Absolute differences between the ABLHs retrieved from high-resolution total attenuated backscatter time series at 1064 nm measured over Potenza on April 21, 2010 and the corresponding ABLHs derived from atmospheric temperature and pressure profiles as provided by ECMWF forecasts. ABLHs retrieved by using ECMWF forecasts data are assumed to be the reference. The original time resolution of ECMWF forecasts data is 1 hour. They have been interpolated at the lidar data time resolution (1 minute). Black, red and yellow lines show the absolute differences with respect to the reference of the ABLHs retrieved by MIPA, derivative and WCT algorithms, respectively.



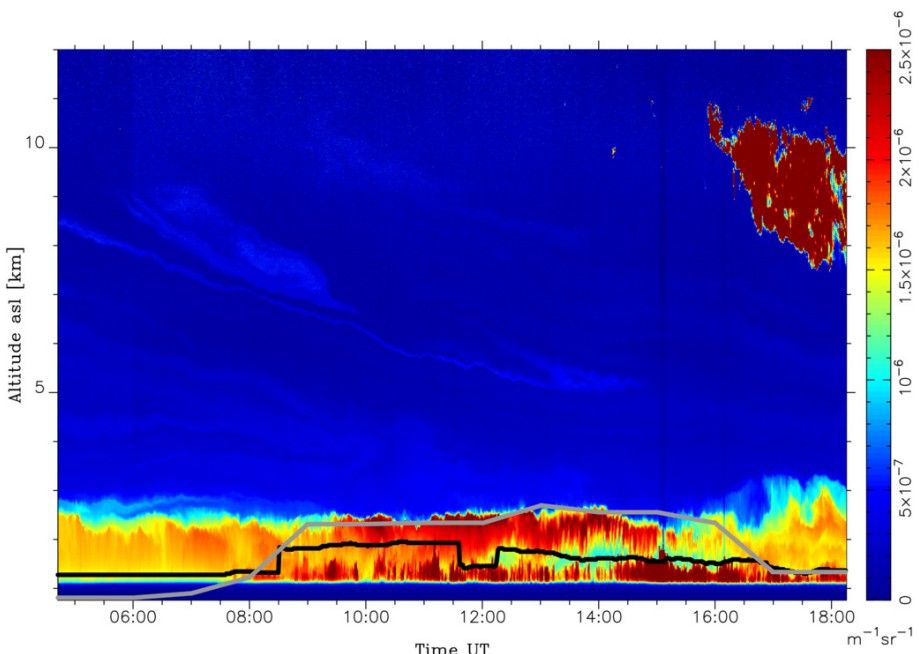

**Figure 11.** High-resolution time series of total attenuated backscatter at 1064 nm measured over Potenza on April 21, 2010. Time resolution is 60 s, vertical resolution is 3.75 m. The black and white curve shows the ABLH as retrieved by MIPA algorithm and by using ECWMF forecasts, respectively.

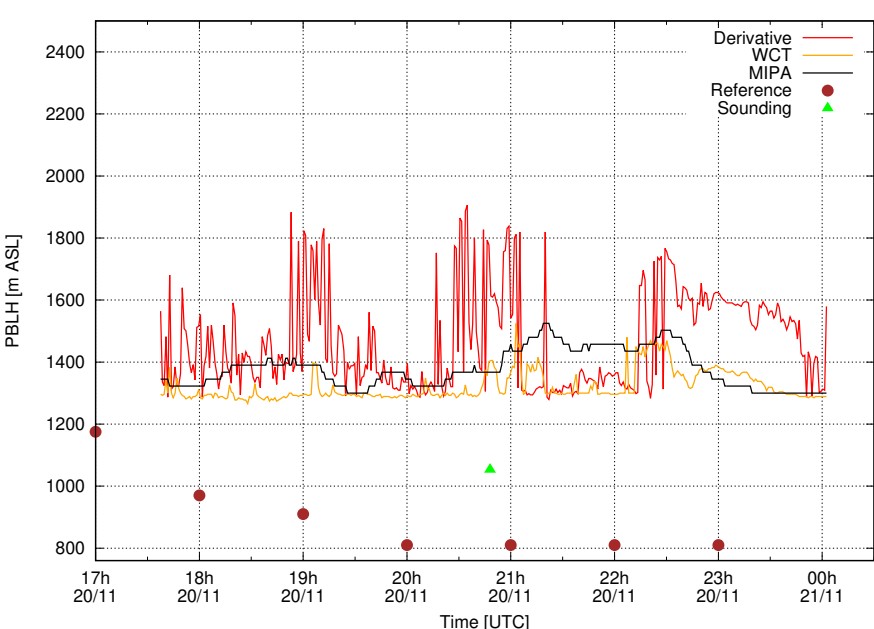

**Figure 12.** ABLH retrieved from the Potenza lidar measurements on November 20, 2014. Black, red and yellow lines show the ABLH retrieved from 1064 nm high-resolution total attenuated backscatter time series applying MIPA, derivative and WCT algorithms, respectively. Brown circles represent the ABLHs as retrieved from atmospheric temperature and pressure profiles provided by ECMWF forecasts. The green triangle represents the ABLH derived from a radiosounding launched at the CIAO observatory.

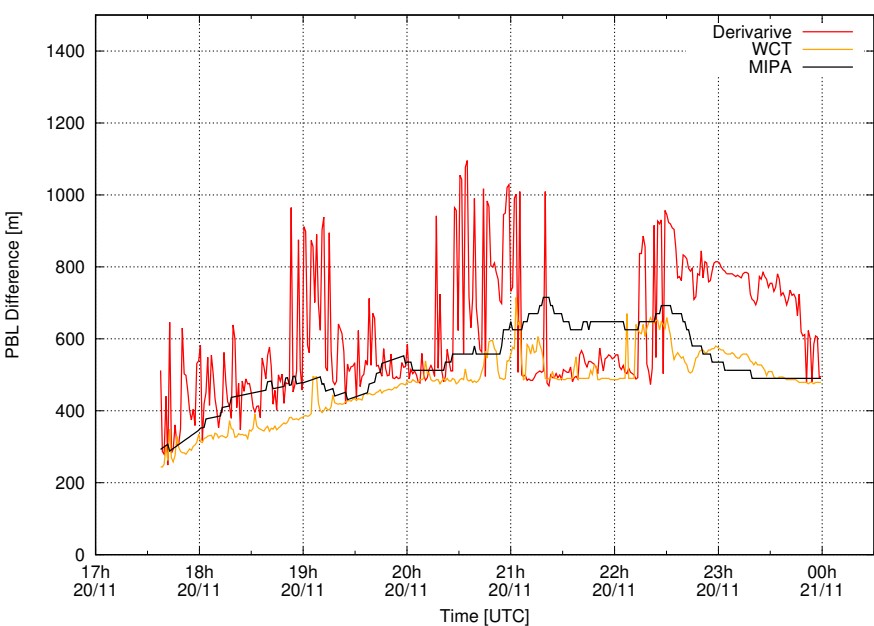

**Figure 13.** Absolute differences between the ABLHs retrieved from high-resolution total attenuated backscatter time series at 1064 nm measured over Potenza on November 20, 2014 and the corresponding ABLHs derived from atmospheric temperature and pressure profiles as provided by ECMWF forecasts. ABLHs retrieved by using ECMWF forecasts data are assumed to be the reference. The original time resolution of ECMWF forecasts data is 1 hour. They have been interpolated at the lidar data time resolution (1 minute). Black, red and yellow lines show the absolute differences with respect to the reference of the ABLHs retrieved by MIPA, derivative and WCT algorithms, respectively.





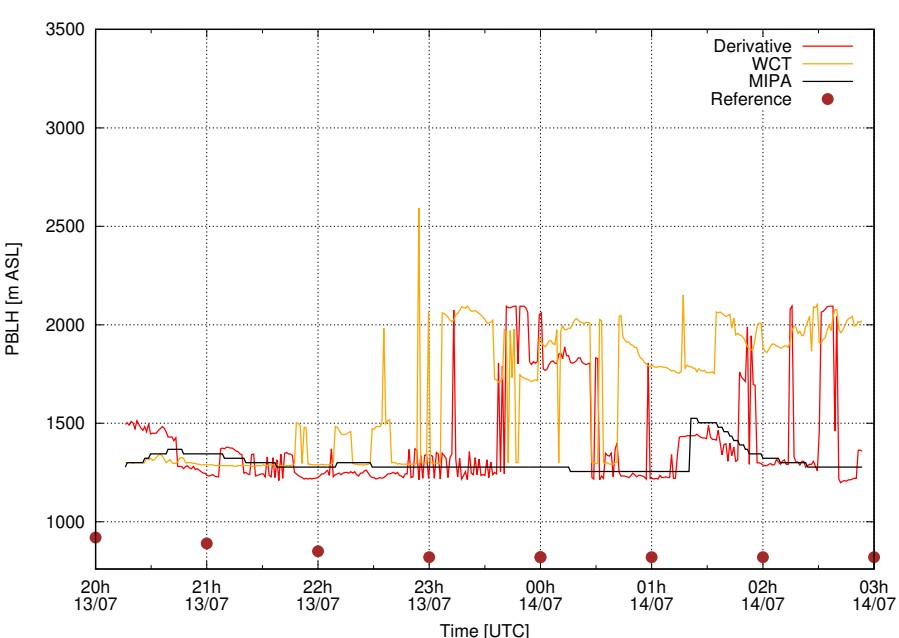

**Figure 14.** ABLH retrieved from the Potenza lidar measurements on July 13, 2015. Black, red and yellow lines show the ABLH retrieved from 1064 nm high-resolution total attenuated backscatter time series applying MIPA, derivative and WCT algorithms, respectively. Brown circles represent the ABLHs as retrieved from atmospheric temperature and pressure profiles provided by ECMWF forecasts.

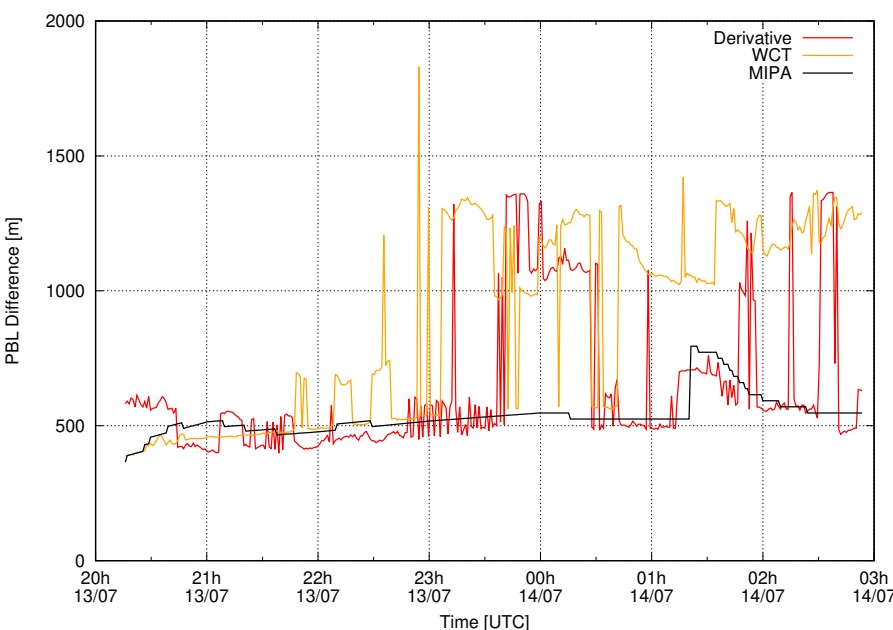

**Figure 15.** Absolute differences between the ABLHs retrieved from high-resolution total attenuated backscatter time series at 1064 nm measured over Potenza on July 13, 2015 and the corresponding ABLHs derived from atmospheric temperature and pressure profiles as provided by ECMWF forecasts. ABLHs retrieved by using ECMWF forecasts data are assumed to be the reference. The original time resolution of ECMWF forecasts data is 1 hour. They have been interpolated at the lidar data time resolution (1 minute). Black, red and yellow lines show the absolute differences with respect to the reference of the ABLHs retrieved by MIPA, derivative and WCT algorithms, respectively.





**Table 4.** Statistical analysis of the absolute differences with respect to the reference of the ABLH retrieved by applying MIPA, WCT and derivative approaches on Potenza high-resolution time series of the total attenuated backscatter at 1064 nm in the 3 selected cases study (April 21, 2010, November 20, 2014, July 13, 2015). The mean ($\Delta_{mean}$), the median ($\Delta_{mean}$), the standard deviation ($\Delta_{std}$), the standard error ($\Delta_{ste}$), the minimum and the maximum of the absolute differences are given in meters. $N$ is the number of points on which the statistics are made. The reference is assumed to be the ABLH calculated from the co-located atmospheric temperature and pressure profiles provided by ECMWF forecasts.

| | MIPA | WCT | Derivative |
|---|---|---|---|
| **2010-04-21** | | | |
| $\Delta_{mean}$ | 519 | 868 | 850 |
| $\Delta_{med}$ | 385 | 871 | 864 |
| $\Delta_{std}$ | 321 | 478 | 439 |
| $\Delta_{ste}$ | 84 | 128 | 117 |
| $\Delta_{min}$ | 24 | 83 | 26 |
| $\Delta_{max}$ | 963 | 1613 | 1420 |
| $N$ | 14 | 14 | 14 |
| **2014-11-20** | | | |
| $\Delta_{mean}$ | 531 | 470 | 610 |
| $\Delta_{med}$ | 543 | 508 | 578 |
| $\Delta_{std}$ | 100 | 93 | 119 |
| $\Delta_{ste}$ | 41 | 38 | 49 |
| $\Delta_{min}$ | 378 | 325 | 450 |
| $\Delta_{max}$ | 645 | 552 | 790 |
| $N$ | 6 | 6 | 6 |
| **2015-07-13** | | | |
| $\Delta_{mean}$ | 466 | 829 | 614 |
| $\Delta_{med}$ | 452 | 928 | 500 |
| $\Delta_{std}$ | 35 | 275 | 236 |
| $\Delta_{ste}$ | 13 | 104 | 89 |
| $\Delta_{min}$ | 437 | 406 | 405 |
| $\Delta_{max}$ | 540 | 1125 | 947 |
| $N$ | 7 | 7 | 7 |