# Peer review of "Atmospheric Boundary Layer height estimation from aerosol lidar: a new approach based on morphological image processing techniques"

_Atmospheric Chemistry and Physics, 2020_

## Referee Comment (RC1) · Anonymous Referee #1 · 21 Oct 2020

The paper entitled "Atmospheric Boundary Layer height estimation from aerosol lidar: a new approach based on morphological image processing techniques" approaches a new methodology to estimate the ABLH from the image/signal processing. The paper is innovative and presents an interesting solution to ABLH detection from elastic lidar/ceilometer data. In the clear sky and convective situations, the algorithm proposed has good results, however in complex or stable situations (e.g. presence of decoupled aerosol layers) the algorithm does not find the same performance.

General Questions:

Performing a separation between the different sublayers of ABLH (Convective Boundary Layer, Residual Layer, Stable Layer, etc.) and then compare how the method behaves when estimating each one can generate better results. Especially during the night.

If Canny's edge detector of other computational languages is applied, is it possible to find the same results? Thinking about the dissemination of this algorithm in other researches centers, an open-source library can be a better solution.

Considering the overlap values of the lidar systems, which is the layer detected by the algorithm during stable situations?

Considering the edge detection, how the specific setup (e.g. energy of the laser beam) of the lidar system can affect the results?

Line 398 - I recommend to add the information about the overlap close to the system description.

Line 406 - I recommend to add the information about the overlap close to the system description.

It would be interesting to present a case with the presence of clouds close to ABLH.

Technical Questions:

I recommend using the same pattern of font style and hour format in the figures.

Fig.1 – Time UTC Fig.6 – Time UTC Fig.11 – Time UTC

---

## Referee Comment (RC2) · Anonymous Referee #2 · 10 Nov 2020

This paper aims to improve the knowledge about the atmospheric boundary layer (ABL) dynamics proposing a new method to retrieve the ABL height (ABLH). The technique used is based on image processing techniques, what is an innovative approach opening new research lines for this application. The manuscript is well structured, allowing for a clear comprehension of the research involved. However, I found some general issues to be addressed, together with some specific comments.

General comments:

Firstly, it is not clear to me what is the definition of ABL used by the authors. The residual layer seems to be part of the ABL sometimes through the manuscript (lines 15-19)

, but in other parts of the body text does not (for example when ECMWF retrievals are presented during nighttime). Therefore, this fact must be clarified and homogeneously used along the article.

Secondly (linked to my previous comment), most of the ABLH retrieving algorithms based on lidar data currently published in the literature suffer from a common drawback: a proper layer attribution. The proposed MIPAS algorithm allows for retrieving something that authors attribute to the ABLH. However, this ABLH is ambiguous because what they are retrieving essentially is a sharp decrease in the attenuated backscatter coefficient profile (or its uncalibrated version, i.e. the lidar range corrected signal). How this is attributed to a specific internal sublayer of the ABL is completely arbitrary without additional information such as turbulence or temperature. In my opinion, MIPA algorithm is a really nice new method with high potentiality but needs to be validated against different techniques involving co-located measurements of temperature (microwave radiometer) or turbulence (Doppler lidar).

Some discussions about these two general comments would hugely enrich this manuscript.

Specific comments:

Lines 15-16: This definition is biased to some extent. There are fundamental meteorological processes not occurring in the lowermost atmosphere, for example related to the hydrological cycle. I recommend removing this sentence.

Line 19: please, avoid to cite unpublished articles.

Lines 20-21: this is a generality. Not always stable layers appear during nighttime and not always mixed, mixing and convective layers appear in daytime. Please, remove the sentence after 'solar cycle'.

Lines 23-26: consider to briefly describe the role of snow as the underlying surface and also the topography as a key element to modify the ABL internal structure.

**ACPD**
Lines 28-30: Please, cite some examples such as:

Lyamani, H., Fernández-Gálvez, J., Pérez-Ramírez, D., Valenzuela, A., Antón, M., Alados, I., Titos, G., Olmo, F.J., Alados-Arboledas, L. Aerosol properties over two urban sites in South Spain during an extended stagnation episode in winter season (2012) Atmospheric Environment, 62, pp. 424-432. DOI: 10.1016/j.atmosenv.2012.08.050.

Lines 42-44: The structure of this sentence is focusing because here the concepts of physical quantity and tracer are mixed up. For example, clouds and aerosols are not atmospheric properties to retrieve ABLH; attenuated backscatter coefficient is the physical quantity here.

Line 48: no, the tracer here is aerosol, backscatter profile is the observed quantity.

Lines 60-61: Please, add also:

Baars, H., A. Ansmann, R. Engelmann, and D. Althausen (2008), Continuous monitoring of the boundary-layer top with lidar, Atmos. Chem. Phys., 8, 7281–7296, doi:10.5194/acp-8-7281-2008.

Granados-Munoz, M.J., Navas-Guzmán, F., Bravo-Aranda, J.A., Guerrero-Rascado, J.L., Lyamani, H., Fernández-Gálvez, J., Alados-Arboledas, L. Automatic determination of the planetary boundary layer height using lidar: One-year analysis over southeastern Spain, (2012) Journal of Geophysical Research Atmospheres, 117 (17), art. no. D18208, DOI: 10.1029/2012JD017524.

Line 90: Consider to replace the term 'tracer'.

Line 101: regarding 'capping temperature inversion'. I guess you are referring to the entraiment zone (region leading to incorporate clean air from the troposphere into the ABL and, thus, to its growth). Capping temperature inversion mainly occurs during nighttime.

Lines 101-103: This sentence is rather pretentious and it is far from being truth in
general. From lidar signals is not possible to unambiguously determine neither the ABL or its internal structure. Essentially, lidar data allows for recognizing layer's edges but the attribution of them to specific internal ABL sublayers is not possible with standalone lidar data. To correctly perform this attribution, complementary information such as stability analysis or turbulence data are needed.

Lines 117-120: This is only valid under scenarios of no decoupled layers. Under the presence of strong layering inside the ABL or decoupled aerosol layers in the free troposphere, the minimum in equation 3 might correspond to a layer or sublayer different from the ABL. See Morille et al (2007) or Granados-Muñoz et al. (2012).

Section 3: Congratulations for this detailed description of the methodology used. Really nice and informative.

Lines 211-212: why is this value of 20 m used?

Lines 248-249: What is the threshold to consider a variation as too fast? I mean, the temporal scale to consider an edge change as unrealistic. Is this threshold the same during the whole daily cycle? This must be taking into account that since faster changes are expected, in general, at sunrise and sunset than during nighttime.

Lines 270-273: It is advisable to transform this subsection into a flux diagram.

Lines 278-280: Thanks to the strict standardization from the instrumental and algorithm point of view, all lidar systems in EARLINET are comparable, therefore this statement makes no sense. For the scope of study, to include different datasets during this 72h-intensive exercise covering different ABL types would enrich this study: topography (mountain versus flat), inland/coastal, latitude dependence, etc. I encourage you to extend the analysis to other relevant stations participating in this intensive exercise to cover as much as possible a variety of atmospheric scenarios.

Lines 283-284: How does the depolarization ratio is performed? No need to provide a large explanation, perhaps you can provide some references on how this is carried
out.

Line 288: Similarly to MUSA, please provide a reference where to see more technical features of PAOLI.

Line 290: replace 'green' by 'visible channel' or '532 nm'.

Line 293: Information about the full overlap height is crucial for all ABLH methods based on vertical changes of the lidar signals. It is mandatory to include such kind of information here.

Line 293-294: Again I consider that the authors focused incorrectly on the technical differences of the lidar systems used. Because of the standardization protocols in EARLINET, both systems should be equivalents. The focus should be moved to the different characteristics of the ABL over both stations.

Line 295: Is this attenuated backscatter coefficient already calibrated or uncalibrated (the latter, therefore, range corrected signal)?

Line 317-321: Some details on the meaning of high resolution should be included here. For example, this resolution is 5 min? 1 min? less than 60 s?

Line 323: replace 'PBL' by 'ABL' (also in other parts of the manuscript).

Lines 325-328: this sentence is repetitive (already presented previously in some way) and, therefore, might be removed.

Lines 329-342: Due to ABL strongly depends on many factors, only co-located and simultaneous radiosoundings must be used for validation of ABLH algorithm comparisons. Because of the lack of such kind of usable radiosoundings, please remove all radiosounding information and data in this article, and focus your validation on NWP data. Another possibility is to compare with microwave radiometer derived profiles.

Lines 357-358: Is the GDAS accuracy the same as the ECWMF NWP? You should use an homogeneous dataset for all the stations involved in the study.

ACPD
Lines 361-362: There are several publications reporting the atmospheric scenarios occurred during the 72-h exercise. Please, consider to cite them in case the readers wish additional details.

Lines 369: replace 'convection' by 'convection'.

Lines 381-390: Please, combine figure 2 and 3 into one figure with two panels in order to better identify the agreements/disagreements. The same as the next pairs of figures. Also, there are huge discrepancies during nighttime and this is not appropriately discussed. In my opinion, the reason is that the model retrieved the top of the stable boundary layer whereas your lidar-based method retrieves the top of the residual layer. Is it possible to get information on the top of the residual layer from model by using the potential temperature or, even better, potential virtual temperature? Likely, the agreement will improve.

Lines 397-398: The information on the full overlap height should have been given when the MUSA technical features were presented.

Lines 400-401: Again, the references are chosen to be the ECMWF retrievals, what during nighttime do not 'see' the same ABL structure than the lidar-based methods. This issue needs to be appropriately discussed in detail.

Lines 409-411: It doesn't make sense to validate an ABL retrieving algorithm using radiosondes launched 110 km far away. To my knowledge Lisbon is a huge city, therefore affected by heat island related effects, whereas Évora is a small, non-industrialized city with very different atmospheric conditions.

Lines 415-417: The overlap is not the only effect, MIPA and EMCWF allow for retrieving different ABL sublayers!

Lines 464-466: Since the computational cost is really low, I encourage to apply the MIPA approach to the whole 72h exercise (all the participant stations) in order to get more robust conclusions.

**ACPD**
Figures 2 and 4: Why does the ECMWF model systematically produce the same ABLH during nighttime (independently on the time along the night, and also the date)?

**ACPD**

---

## Author Comment (AC2) · 10 Jan 2021

**ACP-2020-857, Atmospheric Boundary Layer height estimation from aerosol lidar: a new approach based on morphological image processing techniques**

Authors: Gemine Vivone, Giuseppe D'Amico, Donato Summa, Simone Lolli, Aldo Amodeo, Daniele Bortoli, and Gelsomina Pappalardo

We wish to thank the Associate Editor and all the Reviewers for the thoughtful comments. We appreciate the constructive comments and suggestions, which have helped us improving the quality of the paper. Corrections and updates have been made according to the Reviewers' indications and replies to each concern can be found below.

**Reply to Reviewer 2**

This paper aims to improve the knowledge about the atmospheric boundary layer (ABL) dynamics proposing a new method to retrieve the ABL height (ABLH). The technique used is based on image processing techniques, what is an innovative approach opening new research lines for this application. The manuscript is well structured, allowing for a clear comprehension of the research involved. However, I found some general issues to be addressed, together with some specific comments.

We would like the anonymous Reviewer for considering the approach innovative and the manuscript well-structured.

1. Firstly, it is not clear to me what is the definition of ABL used by the authors. The residual layer seems to be part of the ABL sometimes through the manuscript (lines 15-19), but in other parts of the body text does not (for example when ECMWF retrievals are presented during nighttime). Therefore, this fact must be clarified and homogeneously used along the article.

   We would like to thank the anonymous Reviewer for this comment. We strongly agree with this suggestion. In Sect. 4 (Results, lines 330-343), we have better clarified the ABL definitions that have been assumed, even providing more information on the procedure for the calculation of the ABLH using NWP.

2. Secondly (linked to my previous comment), most of the ABLH retrieving algorithms based on lidar data currently published in the literature suffer from a common drawback: a proper layer attribution. The proposed MIPAS algorithm allows for retrieving something that authors attribute to the ABLH. However, this ABLH is ambiguous because what they are retrieving essentially is a sharp decrease in the attenuated backscatter coefficient profile (or its uncalibrated version, i.e. the lidar range corrected signal). How this is attributed to a specific internal sublayer of the ABL is completely arbitrary without additional information such as turbulence or temperature. In my opinion, MIPA algorithm is a really nice new method with high potentiality but needs to be validated against different techniques involving co-located measurements of temperature (microwave radiometer) or turbulence (Doppler lidar).

   We strongly agree with the anonymous Reviewer's suggestion. Sect. 4 (Results) has been modified mentioning the important point raised by the Reviewer.

   The use of temperature profiles retrieved by microwave radiometers is under investigation. However, it is important to note that microwave radiometers do not directly measure

temperature profiles, but they derive them from brightness temperature measurements applying not straightforward retrieval algorithms (e.g., optimal estimation approaches or neural networks). Consequently, it is not ensured that temperature inversions (which play a crucial role for the ABLH detection) are always well captured. Concerning the turbulence study, unfortunately, there are no co-located Doppler lidar data available for any of the test cases considered in the paper.

Finally, the authors fully agree with the Reviewer in thinking that a better validation of MIPA is needed and, indeed, they have planned to prepare another paper in which this important topic is carefully addressed considering (among other things) also what the Reviewer suggested. The aim of the present work is just to describe the novel approach providing a proof of concept, i.e., comparing the obtained results with some consolidated lidar-based techniques with a "basic" validation. The authors think that a deeper validation of MIPA is behind the scope of this work and it is more appropriate to address it in a dedicated paper because there are many aspects to be considered (synergies with different instruments as suggested by the Reviewer, evaluation of better reference based on mesoscale models instead of global ones, the use of other lidar wavelenghts, the consideration of more test cases acquired by other instruments and in other locations, and so forth).

Some discussions about these two general comments would hugely enrich this manuscript.

Specific comments:

1. Lines 15-16: This definition is biased to some extent. There are fundamental meteorological processes not occurring in the lowermost atmosphere, for example related to the hydrological cycle. I recommend removing this sentence.
   Done. Thanks.

2. Line 19: please, avoid to cite unpublished articles.
   Removed. Thanks.

3. Lines 20-21: this is a generality. Not always stable layers appear during nighttime and not always mixed, mixing and convective layers appear in daytime. Please, remove the sentence after 'solar cycle'.
   Changed accordingly. Thanks

4. Lines 23-26: consider to briefly describe the role of snow as the underlying surface and also the topography as a key element to modify the ABL internal structure.
   We thank the referee for the meaningful comment. Of course the underlying surface is playing a fundamental role in boundary layer development. However, the scope of this manuscript is to provide a proof of concept on how our algorithm works. We will take into consideration this remark for future developments. A clarification has been added in the text.

5. Lines 28-30: Please, cite some examples such as:
   Lyamani, H., Fern ndez-G lvez, J., P rez-Ram rez, D., Valenzuela, A., Ant n, M., Alados, I., Titos, G., Olmo, F.J., Alados-Arboledas, L. Aerosol properties over two urban sites in South Spain during an extended stagnation episode in winter season (2012) Atmospheric Environment, 62, pp. 424-432. DOI: 10.1016/j.atmosenv.2012.08.050.
   Added

6. Lines 42-44: The structure of this sentence is focusing because here the concepts of physical quantity and tracer are mixed up. For example, clouds and aerosols are not atmospheric properties to retrieve ABLH; attenuated backscatter coefficient is the physical quantity

here.

*We changed the sentence accordingly. Thanks.*

7. Line 48: no, the tracer here is aerosol, backscatter profile is the observed quantity.

   *Changed accordingly. Thanks.*

8. Lines 60-61: Please, add also:

   Baars, H., A. Ansmann, R. Engelmann, and D. Althausen (2008), Continuous monitoring of the boundary ?AR? layer top with lidar, Atmos. Chem. Phys., 8, 7281  7296, doi:10.5194/acp ?A ? R8 ?A ? R7281 ?AR? 2008.

   Granados-Munoz, M.J., Navas-Guzm n, F., Bravo-Aranda, J.A., Guerrero-Rascado, J.L., Lyamani, H., Fern ndez-G lvez, J., Alados-Arboledas, L. Automatic determination of the planetary boundary layer height using lidar: One-year analysis over southeastern Spain, (2012) Journal of Geophysical Research Atmospheres, 117 (17), art. no. D18208, DOI: 10.1029/2012JD017524.

   *References added. Thanks.*

9. Line 90: Consider to replace the term 'tracer'.

   *The sentence has been rephrased avoiding the word 'tracer'. Thanks for your suggestion.*

10. Line 101: regarding 'capping temperature inversion'. I guess you are referring to the entraiment zone (region leading to incorporate clean air from the troposphere into the ABL and, thus, to its growth). Capping temperature inversion mainly occurs during nighttime.

    *Done. Thanks.*

11. Lines 101-103: This sentence is rather pretentious and it is far from being truth in general. From lidar signals is not possible to unambiguously determine neither the ABL or its internal structure. Essentially, lidar data allows for recognizing layer's edges but the attribution of them to specific internal ABL sublayers is not possible with standalone lidar data. To correctly perform this attribution, complementary information such as stability analysis or turbulence data are needed.

    *The sentence has been rephrased. Thanks for your suggestion.*

12. Lines 117-120: This is only valid under scenarios of no decoupled layers. Under the presence of strong layering inside the ABL or decoupled aerosol layers in the free troposphere, the minimum in equation 3 might correspond to a layer or sublayer different from the ABL. See Morille et al (2007) or Granados-Mu oz et al. (2012).

    *Thanks for the comment. This latter has been included in the revised version of the paper.*

13. Section 3: Congratulations for this detailed description of the methodology used. Really nice and informative.

    *We would like to thank the anonymous Reviewer for the congratulations about Sect. 3.*

14. Lines 211-212: why is this value of 20 m used?

    *This value has been empirically set in order to avoid multiple edges corresponding to the same layer. Thus, we can have sharper and uniquely identifiable edges. This aspect has been clarified in the revised Sect. 3.2.*

15. Lines 248-249: What is the threshold to consider a variation as too fast? I mean, the temporal scale to consider an edge change as unrealistic. Is this threshold the same during the whole daily cycle? This must be taking into account that since faster changes are expected, in general, at sunrise and sunset than during nighttime.

    *Thanks for the comment. We do not consider any threshold to discriminate a variation as *fast* or *non-fast*. Instead, we apply a series of directional low-pass morphological filters*

as post-processing to pursue the same goal. Obviously, the length and the maximum and minimum angles of these filters rule the filtering of these edges, but there is no direct link with the fast variation detection as in the case of the application of a threshold. About the second comment, we think this is really interesting. Indeed, as underlined by the anonymous Reviewer, this analysis could be improved by applying different post-processing filters along the whole daily cycle in order to consider the sunrise/sunset in a different way with respect to the rest of the day. These will surely increase the performance but paying it with the introduction of an *a priori* information about the daily cycle getting an algorithm that is not completely blind. Anyway, the introduction of physical *a priori* information to improve the performance deserves future developments.

Lines 270-273: It is advisable to transform this subsection into a flux diagram.
Thanks for the comment. A flowchart has been added in Sect. 3.6.

16. Lines 278-280: Thanks to the strict standardization from the instrumental and algorithm point of view, all lidar systems in EARLINET are comparable, therefore this statement makes no sense. For the scope of study, to include different datasets during this 72h intensive exercise covering different ABL types would enrich this study: topography (mountain versus flat), inland/coastal, latitude dependence, etc. I encourage you to extend the analysis to other relevant stations participating in this intensive exercise to cover as much as possible a variety of atmospheric scenarios.
    In general, the authors agree with the Reviewer concerning the EARLINET standardization statement. However, the EARLINET standardization does not apply too much to the lidar instruments that actually can be very different one from the other. In EARLINET there are commercial, home-made and hybrid instruments that can differ in many important aspects like laser type and model, telescope design (some lidars, for example, operate with different telescopes, some others with only one), detectors types (PMT, APD), acquisition systems All these differences in the hardware are reflected even in the EARLINET products. In general, the more advanced the EALRINET product, the more the level of standardization. For instance, the level of standardization is quite high on aerosol optical products (aerosol extinction, backscatter and depolarization ratio profiles) but typically lower in the more basic products. In this study we have used as input the EARLINET high-resolution pre-processed products (total attenated backscatter timeseries) in which some instrumental differences can have an important impact (e.g., different spatial and temporal resolutions). For this reason, the authors would prefer to mention the good applicability of MIPA for different lidar instruments because they consider it an important advantage of the proposed methodology. Nevertheless, the authors recognize that the Reviewer is right in pointing out also the difference in terms of topography of the two measurements sites. The sentence has been modified to stress even this aspect.

17. Lines 283-284: How does the depolarization ratio is performed? No need to provide a large explanation, perhaps you can provide some references on how this is carried out.
    The authors did not provide more details on how the depolarization ratio is performed because this quantity and the corresponding lidar channels are never used in this work (we only used the total attenuated backscatter at 1064nm). Anyway, the depolarization ratio is calculated according to the standard EARLINET protocol: the calibration is made by using $\Delta 90$ method and other experimental effects (like cross-talk, diattenuation, …) are taken into account by specific parameters calculated for cross and parallel polarization channels ($GHk$ parameters). All the details about this procedure are described by Freudenthaler, V., Atmos. Meas. Tech., 9, 4181-4255, https://doi.org/10.5194/amt-9-4181-2016. This reference has been added in the manuscript.

18. Line 288: Similarly to MUSA, please provide a reference where to see more technical features of PAOLI.

Done. Thanks.

19. LIne 290: replace 'green' by 'visible channel' or '532 nm'.
Done. Thanks.

20. Line 293: Information about the full overlap height is crucial for all ABLH methods based on vertical changes of the lidar signals. It is mandatory to include such kind of information here.
Done. Thanks.

21. Line 293-294: Again I consider that the authors focused incorrectly on the technical differences of the lidar systems used. Because of the standardization protocols in EARLINET, both systems should be equivalents. The focus should be moved to the different characteristics of the ABL over both stations.
As for the previous comment on the same topic, the authors think it could be important to underline the differences between Potenza and Evora lidars because they can play a role in the specific products we used as input for the MIPA algorithm. Nevertheless, we have taken into account the Reviewer suggestion also mentioning the different characteristics of the two measurement sites.

22. Line 295: Is this attenuated backscatter coefficient already calibrated or uncalibrated (the latter, therefore, range corrected signal)?
In this work we always applied MIPA algorithm on the total attenuated backscatter timeseries at 1064nm, which is a calibrated quantity (basically it is the total bacskcatter coefficient attenuated for atmospheric transmissivity). Nevertheless, one of the main strength of the proposed approach is that it is quite insensitive to the absolute values of the input dataset. In other words, the edge detection and the morphological filters rely only on the correlations (in both time and space) of neighbor pixels in input. Consequently, very similar results can be obtained if MIPA is applied on range corrected signal timeseries using the same algorithm configuration settings used for the total attenuated backscatter (the ones summarized in Tab. 1).

23. LIne 317-321: Some details on the meaning of high resolution should be included here. For example, this resolution is 5 min? 1 min? less than 60 s?
The sentence has been re-phrased according to the Reviewer suggestion.

24. Line 323: replace 'PBL' by 'ABL' (also in other parts of the manuscript).
Done. Thanks.

25. Lines 325-328: this sentence is repetitive (already presented previously in some way) and, therefore, might be removed.
Done. Thanks.

26. Lines 329-342: Due to ABL strongly depends on many factors, only co-located and simultaneous radiosoundings must be used for validation of ABLH algorithm comparisons. Because of the lack of such kind of usable radiosoundings, please remove all radiosounding information and data in this article, and focus your validation on NWP data. Another possibility is to compare with microwave radiometer derived profiles.
Actually the few sounding data available were shown only with the intention to corroborate the authors choice to use as reference the ABLH retrieved from NWP forecasts. They were never used as reference in the article. Anyway, we recognize this can be somehow confusing and, following the Reviewer suggestion, we have reshaped the sentence and removed all the occurrences about radiosounding that can be found in the paper. Finally, the use of temperature profiles retrieved by microwave radiometers is under investigation. Anyway, it is important to note that microwave radiometers do not directly measure temperature

profiles, but they derive them from brightness temperature measurements applying not straightforward retrieval algorithms (like, e.g., optimal estimation approaches or neural networks). Consequently, it is not ensured that temperature inversions (which play a crucial role for the ABLH detection) are always well captured.

27. Lines 357-358: Is the GDAS accuracy the same as the ECWMF NWP? You should use an homogeneous dataset for all the stations involved in the study.
The authors agree with the Reviewer but, unfortunately, they have no access to the high resolution ECMWF NWP for Evora during the 3 days of 72h exercise. Thus, for the Evora case study, the only possibility is to use GDAS NWP data, which are freely available. Anyway, the main differences between GDAS and ECMWF NWP have been briefly described at the end of Sect. 4 (Results) in a way that the reader can figure out the consequences of using GDAS instead of ECMWF NWP.

28. LInes 361-362: There are several publications reporting the atmospheric scenarios occurred during the 72-h exercise. Please, consider to cite them in case the readers wish additional details.
The authors thanks the Reviewer for this suggestion. The following references have been added:

   (a) Wang, Y., at al: Assimilation of lidar signals: application to aerosol forecasting in the western Mediterranean basin, Atmos. Chem. Phys., 14, 1203-12053, https://doi.org/10.5194/acp-14-12031-2014, 2014.
   (b) Sicard, M. et al.: EARLINET: potential operationality of a research network, Atmos. Meas. Tech., 8, 4587-4613, https://doi.org/10.5194/amt-8-4587-2015, 2015.
   (c) Granados-Muñoz, M. J., at all: Profiling of aerosol microphysical properties at several EARLINET/AERONET sites during the July 2012 ChArMEx/EMEP campaign, Atmos. Chem. Phys., 16, 7043-7066, https://doi.org/10.5194/acp-16-7043-2016, 2016.
   (d) M. Sicard, et al.: Contribution of EARLINET/ACTRIS to the summer 2013 Special Observing Period of the ChArMEx project: monitoring of a Saharan dust event over the western and central Mediterranean, International Journal of Remote Sensing, 37:19, 4698-4711, DOI: 10.1080/01431161.2016.1222102.

29. Lines 369: replace 'convection' by 'convection'.
Done. Thanks.

30. Lines 381-390: Please, combine figure 2 and 3 into one figure with two panels in order to better identify the agreements/disagreements. The same as the next pairs of figures. Also, there are huge discrepancies during nighttime and this is not appropriately discussed. In my opinion, the reason is that the model retrieved the top of the stable boundary layer whereas your lidar-based method retrieves the top of the residual layer. Is it possible to get information on the top of the residual layer from model by using the potential temperature or, even better, potential virtual temperature? Likely, the agreement will improve.
The authors thanks the Reviewer for the useful comment. Following the Reviewer suggestion, the following figures have been combined: figures 2 and 3; figures 4 and 5; figures 7 and 8; figures 9 and 10; figures 12 and 13; figures 14 and 15. Moreover, concerning the discrepancies, more explanations have provided in the manuscripts, accordingly.

31. Lines 397-398: The information on the full overlap height should have been given when the MUSA technical features were presented.
Done. Thanks.

32. Lines 400-401: Again, the references are chosen to be the ECMWF retrievals, what during nighttime do not 'see' the same ABL structure than the lidar-based methods. This issue needs to be appropriately discussed in detail.
The authors fully agree with the Reviewer. More explanations have been provided.

33. Lines 409-411: It doesn't make sense to validate an ABL retrieving algorithm using radiosondes launched 110 km far away. To my knowledge Lisbon is a huge city, therefore affected by heat island related effects, whereas Evora is a small, non-industrialized city with very different atmospheric conditions.
According to the Reviewer suggestion, the part involving the sounding data has been removed.

34. Lines 415-417: The overlap is not the only effect, MIPA and EMCWF allow for retrieving different ABL sublayers!
The authors fully agree with the Reviewer. More explanations have been provided.

35. Lines 464-466: Since the computational cost is really low, I encourage to apply the MIPA approach to the whole 72h exercise (all the participant stations) in order to get more robust conclusions.
Thanks for the comment. However, we think that this is out of scope for this work. The main goal of the authors is to provide a proof of concept about a new approach to derive the ABLH from lidar observations. The new approach was applied on several cases studies corresponding to different atmospheric conditions and also to different lidar systems. In our opinion, the number of test cases included in the paper can be considered enough considering the authors' scope. Moreover, the authors have already planned to perform further studies on this topic to provide a more solid and robust validation of the proposed MIPA algorithm. In these studies, all the 72h exercise dataset will be considered.

36. Figures 2 and 4: Why does the ECMWF model systematically produce the same ABLH during nighttime (independently on the time along the night, and also the date)?
This is because during nighttime there were very stable atmospheric conditions over Potenza for all the 3 days of the 72h exercise. In particular, the nighttime ABLH we derived applying the procedure described in [1] on the potential temperature profiles provided by ECMWF NWP are quite stable and low (below 100 m). Such low and stable values can be explained by considering that in nighttime conditions the procedure we have used gives an estimation of the stable layer top which under calm conditions (low wind) and with small surface cooling can be considerably low. This seems to be the case as in Potenza for all the 72h exercise nights, where between 20:00UT and 05:00UT at surface we measured a mean temperature of 19 °C and an average wind speed of 0.9 m/s (with a standard deviation of 0.9 m/s).

[1] Liu, S. and X.-Z. Liang, 2010: Observed diurnal cycle climatology of planetary boundary layer height. J. Climate, 23, 5790 5809.

Sincerely,
The Authors

---

## Author Comment (AC1)

**ACP-2020-857, Atmospheric Boundary Layer height estimation from aerosol lidar: a new approach based on morphological image processing techniques**

**Authors: Gemine Vivone, Giuseppe D'Amico, Donato Summa, Simone Lolli, Aldo Amodeo, Daniele Bortoli, and Gelsomina Pappalardo**

We wish to thank the Associate Editor and all the Reviewers for the thoughtful comments. We appreciate the constructive comments and suggestions, which have helped us improving the quality of the paper. Corrections and updates have been made according to the Reviewers' indications and replies to each concern can be found below.

**Reply to Reviewer 1**

The paper entitled "Atmospheric Boundary Layer height estimation from aerosol lidar: a new approach based on morphological image processing techniques" approaches a new methodology to estimate the ABLH from the image/signal processing. The paper is innovative and presents an interesting solution to ABLH detection from elastic lidar/ceilometer data. In the clear sky and convective situations, the algorithm proposed has good results, however in complex or stable situations (e.g. presence of decoupled aerosol layers) the algorithm does not find the same performance.

We would like to thank the anonymous Reviewer for considering our paper innovative and the solution interesting.

1. Performing a separation between the different sublayers of ABLH (Convective Boundary Layer, Residual Layer, Stable Layer, etc.) and then compare how the method behaves when estimating each one can generate better results. Especially during the night.

   We agree with the reviewer that this section is not enough clear. Our methodology uses aerosols as proxy to retrieve the atmospheric boundary layer height, while classic methods, e.g. radiosoundes, use the atmosphere thermodynamics , i.e. retrieving the atmospheric boundary layer from the potential temperature profiles. While during convective boundary layers the two methods are equivalent, at night, they are not because from the aerosol point of view the boundary layer collapses into a stable layer and a residual layer. For this reason, during nighttime, it is difficult to perform an intercomparison because the methodologies are different. Because our algorithm detects edges, we set it up to detect the first edge at night because it is more close to the thermodynamic detection. Problems of course arise when the boundary layer height falls below the lidar overlap region.

2. If Canny's edge detector of other computational languages is applied, is it possible to find the same results? Thinking about the dissemination of this algorithm in other researches centers, an open-source library can be a better solution.

   We have already stated in the paper (see Sect. 3.4) that every edge detector can be exploited to extract a first estimation of the ABLH. Thus, the proposed approach is flexible and the edge detection block can be changed to possibly improve the results. Approaches as wavelet covariance transform or gradient-based could be adopted. However, the analysis of the performance varying these strategies in the proposed framework is out-of-the-scope of this paper. About the Reviewer's comment, all the above-mentioned edge detection approaches can be adopted to have a easier dissemination. Even other software platforms

have their own implementation of Canny's edge detector (see, e.g., Python), which can be exploited to have comparable results with the ones in the paper.

3. Considering the overlap values of the lidar systems, which is the layer detected by the algorithm during stable situations?
As explained in the first comment, because the algorithm is setup to detect the first significant edge, the algorithm is sensed to detect the stable layer, unless it falls below the overlap region. In that case, the residual layer is detected.

4. Considering the edge detection, how the specific setup (e.g. energy of the laser beam) of the lidar system can affect the results?
The detection is performed on the retrieved lidar attenuated backscattering, a variable depending just on the atmospheric variables, e.g., aerosol backscattering and extinction coefficients. The retrieval is not directly affected from the laser beam energy. Of course more energy is linked to a higher signal-to-noise ratio. In case of low power lidar systems, e.g., ceilometers, the boundary layer height detection in strong aerosol loading, e.g. during a haze event, could be difficult.

5. Line 398 - I recommend to add the information about the overlap close to the system description.
Line 406 - I recommend to add the information about the overlap close to the system description.
Done. Thanks.

6. It would be interesting to present a case with the presence of clouds close to ABLH.
As our method is based on edge detection, clouds enhance the detection because, being optically thicker than aerosols, in a convective boundary layer, their base is coincident with boundary layer height. In case of clouds above the boundary layer height, the first edge is still detected. Clouds inside the boundary layer is instead a critical situation, but highly unlikely.

7. I recommend using the same pattern of font style and hour format in the figures.
Fig.1   Time UTC Fig.6   Time UTC Fig.11   Time UTC
Done. Thanks.

Sincerely,
The Authors

---

## Author Response (AR2)

**ACP-2020-857, Atmospheric Boundary Layer height estimation from aerosol lidar: a new approach based on morphological image processing techniques**

**Authors: Gemine Vivone, Giuseppe D'Amico, Donato Summa, Simone Lolli, Aldo Amodeo, Daniele Bortoli, and Gelsomina Pappalardo**

The authors have conducted practically all suggested changes. There still are very minor changes I would humbly ask them to make and then the paper will ready to publish without the need to pass by the referees again.

We wish to thank the Associate Editor and all the Reviewers for the thoughtful comments. We appreciate the constructive comments and suggestions, which have helped us improving the quality of the paper. Corrections and updates have been made according to the Reviewers' indications and replies to each concern can be found below.

**Reply to Reviewer 1**

The authors have correctly addressed most of the issues raised by the reviewer's. I recommend the publication of this article after conducting this last issue:
Thanks for this further note.

1. Lines 323-324: the equation to calculate the potential temperature is correct, however the value of the numerator (parameter Ps0) is not correct. Ps0 is the atmospheric pressure at the reference level of 1000 hPa, no the atmospheric standard pressure (1013.25 hPa).
   Done.

2. Also replace 'kelvin degrees' by 'Kelvin'.
   Done.

**Reply to Reviewer 2**

1. In the main document the references, sections, and figures names as indicated as "???". It is necessary to correct it.
   Done.

2. In figures 2, 5, and 8 the units of the color bar are not correctly positioned.
   Done.

3. In Figure 8 there is not a white line.
   Done.

4. I recommend adding legend in figures 2, 5, and 8.
   Done.

Sincerely,
The Authors